# H-SPLID: HSIC-based Saliency Preserving Latent Information Decomposition

**Lukas Miklautz**[1, †, *]   **Chengzhi Shi**[2, *]   **Andrii Shkabrii**[3,4, *]   **Theodoros Thirimachos Davarakis**[2]
**Prudence Lam**[2]   **Claudia Plant**[3, 5, ‡]   **Jennifer Dy**[2, ‡]   **Stratis Ioannidis**[2, ‡]

[1]Department of Machine Learning and Systems Biology, Max Planck Institute
of Biochemistry, Martinsried, Germany  [2]Northeastern University, Boston, MA, USA
[3]Faculty of Computer Science, University of Vienna, Vienna, Austria
[4]Doctoral School Computer Science, University of Vienna, Vienna, Austria
[5]Research Network Data Science, University of Vienna, Vienna, Austria

## Abstract

We introduce H-SPLID, a novel algorithm for learning salient feature representations through the explicit decomposition of salient and non-salient features into separate spaces. We show that H-SPLID promotes learning low-dimensional, task-relevant features. We prove that the expected prediction deviation under input perturbations is upper-bounded by the dimension of the salient subspace and the Hilbert-Schmidt Independence Criterion (HSIC) between inputs and representations. This establishes a link between robustness and latent representation compression in terms of the dimensionality and information preserved. Empirical evaluations on image classification tasks show that models trained with H-SPLID primarily rely on salient input components, as indicated by reduced sensitivity to perturbations affecting non-salient features, such as image backgrounds.

## 1   Introduction

The acquisition of salient, task-relevant features from high-dimensional inputs constitutes a fundamental challenge in representation learning. Such features offer multiple advantages, including reduced dimensionality [2], enhanced generalization and transferability [42, 30], and improved robustness [22, 12]. Nevertheless, learning true salient features remains challenging, as many neural networks operate within a single, entangled latent space that mixes task-relevant signals with redundant information [6, 38]. We illustrate this sensitivity using a simple diagnostic test in Figure 1: a classifier trained to predict the left digit in an image of double digits should ignore perturbations to the right digit, which is irrelevant to the label. However, in practice, we observe that neural networks with high test classification accuracy on the left digits exhibit a significant performance drop when subjected to a high-magnitude adversarial PGD [34] attack ($\epsilon = 1.0$) on the right digits, revealing their dependence on irrelevant, non-salient features. This corroborates several empirical and theoretical studies [3, 36, 19] showing that redundant dimensions enhance vulnerability to attacks. Driven by these findings, we introduce H-SPLID (HSIC-based Saliency-Preserving Latent Information Decomposition), a new method that learns salient features by explicitly decomposing the latent space into two subspaces coupled with information compression regularization: a low-dimensional *salient space*, which contains features essential for classification, and a *non-salient space*, which captures the remaining input variability. Training the same neural network as above with H-SPLID alleviates the dependence on irrelevant features to a large degree, as shown on the right side of Figure 1. Crucially, H-SPLID is significantly less sensitive to right digit perturbations, without any prior knowledge of the redundant region or adversarial training.

---

[*]Equal contribution. [‡]Shared supervision. [†]Main work done during a research stay at Northeastern University.

39th Conference on Neural Information Processing Systems (NeurIPS 2025).

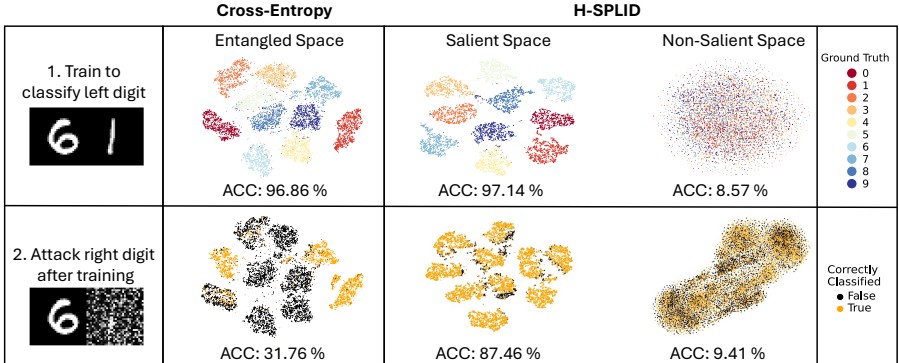

Figure 1: **H-SPLID learns to ignore irrelevant input by decomposing the latent space into salient and non-salient components. Left:** A simple diagnostic test for saliency, where the model is trained to classify the left digit (only labels for the left are provided) and it should ignore the right. **Middle:** A model trained with *cross-entropy* loss achieves high test accuracy (96.86%) but produces entangled representations, making it sensitive to perturbations on the right digit (accuracy drops to 31.76% under high-magnitude PGD attack). **Right:** H-SPLID separates the latent space into a salient subspace, which captures class-discriminative structure (ACC 97.14%), and a non-salient subspace, which contains no class-relevant information (ACC 8.57%). This separation enables robustness to perturbations on irrelevant input (ACC 87.46%), showing proper learning of salient features. Embeddings are visualized using t-SNE [53].

We extend this analysis in our experiments by applying adversarial attacks to the background of COCO [28] images, applying image corruptions to medical images of skin lesions [9] and further show that H-SPLID improves the transfer accuracy of ResNet-based [20] classifiers trained on ImageNet [11] under real-world perturbations [58, 55]. In addition to our empirical evidence, we theoretically prove that the expected change in predictions under input perturbations is bounded by the dimension of the learned salient subspace and the Hilbert-Schmidt Independence Criterion (HSIC) [16] between inputs and salient representations. This establishes a formal link between robustness and the salient representation. Our main contributions are:

- We propose H-SPLID, a novel algorithm that promotes the learning of salient features by decomposing the network's latent space into salient and non-salient subspaces.

- We prove that the two key design components of H-SPLID, namely, dimensionality reduction of the salient subspace, together with the HSIC between inputs and salient latent representations, upper bounds the expected change in predictions under input perturbations. Moreover, we show that the above HSIC and reduced salient space dimensionality bounds the volume of the input domain that is vulnerable to perturbations.

- We empirically demonstrate that H-SPLID learns salient features by leveraging attacks and other perturbations against non-salient regions of an image, such as its background.

## 2   Related Work

**Salient Feature Learning.**   Saliency methods in interpretability aim to identify input features that influence a model's prediction the most. Traditional post hoc approaches include gradient-based methods [44, 48], Class Activation Maps (CAMs) [43, 54], and perturbation-based methods [31, 41]. Unlike post hoc interpretability methods, however, H-SPLID aims to learn latent salient features for a given task, such as image classification. Existing works on salient feature learning include saliency-guided training for interpretability [22] and saliency-based data augmentation [8, 5, 52] methods that can complement our approach. However, H-SPLID does not use saliency maps as an auxiliary signal to improve training [8, 52], or pretrained models [5] to generate them. Moreover, the division of the latent space into "salient" and "non-salient" spaces, as in H-SPLID, is comparatively unexplored in literature. Contrastive Analysis (CA) methods [2, 1, 57] leverage this concept by learning explicit "common" and "salient" latent spaces with separate encoders via external supervision. While they

share the idea of learning separate spaces with H-SPLID, they rely on a dedicated target dataset containing the salient class, and a background dataset with samples exhibiting non-salient features. In contrast, our method does not rely on external data, and learns an initial unified latent space, before partitioning it into salient and non-salient dimensions.

**Feature Decomposition and Selection.** Feature selection methods aim to identify a subset of input features that is most predictive of the target variable [61, 18], with popular approaches including $L_1$ regularization [49] and *Group-Lasso* regularization [62]. Similar to these methods, H-SPLID transforms and selects features during training, but diverges in its approach through its decomposition of the latent space. For the task of clustering, Miklautz et al. [37] recently introduced the idea of latent space partitioning, whereas H-SPLID embeds this split directly into a classifier's training loop, using labels to shape the salient vs. non-salient space. Moreover, H-SPLID incorporates the HSIC penalty [32] to regularize the statistical dependence between the inputs and salient features from each subspace, ensuring that the salient subspace retains only task-relevant information while the non-salient subspace absorbs redundant variability, thereby reducing feature dimensions.

**Adversarial Robustness and Saliency.** We use adversarial attacks to evaluate the quality of the learned salient features. Several studies have begun exploring the interplay between saliency and robustness [12, 51, 17, 29]. Among these methods, many require adversarial training [34], which is not only computationally demanding, but also tailored to specific attacks. An alternative line of research seeks to enhance robustness without adversarial training. Multiple works [19, 3, 56, 13] have attributed adversarial vulnerability to the network's reliance on high-dimensional, task-irrelevant features. For instance, Alemi et al. [3] hypothesize that neural networks falsely rely on task-irrelevant features from the training data, negatively impacting robust generalization. Melamed et al. [36] show, under a simplified two-layer model, that when data is confined to a low-dimensional manifold, there exists an off-manifold space in which weights remain mostly unchanged and can be exploited by adversarial perturbations. Haldar et al. [19] demonstrate that when there are redundant latent dimensions, off-manifold attacks can lead to decision boundaries that rely on task-irrelevant feature dimensions. Fischer [13] introduced an information bottleneck [50] to compress input information and preserve task-relevant features without adversarial training. Wang et al. [56] extends this framework to the HSIC bottleneck [32], upon which H-SPLID is built. Importantly, H-SPLID departs from the HSIC bottleneck penalty, and introduces separate terms for salient and non-salient features. Our method also improves upon the guarantees of Wang et al. [56], tightening their bounds to account for the impact of the dimensionality reduction induced by H-SPLID.

## 3 Methodology

### 3.1 Problem Setup

We consider $k$-class classification over dataset $\mathcal{D} = \{(\mathbf{x}_i, \mathbf{y}_i)\}_{i=1}^n \subseteq \mathcal{X} \times \mathcal{Y}$, where $\mathcal{X} \subseteq \mathbb{R}^d$ is a compact input space, $\mathcal{Y} \subseteq \mathbb{R}^k$ is the label space, and each input $\mathbf{x}_i \in \mathcal{X}$ is associated with the corresponding one-hot class label $\mathbf{y}_i \in \mathcal{Y}$. The neural network $h_{\boldsymbol{\theta}} : \mathbb{R}^d \to \mathbb{R}^k$ consists of an encoder followed by a linear layer. The encoder, denoted by $f_{\boldsymbol{\psi}} : \mathcal{X} \to \mathcal{R}$ with parameters $\boldsymbol{\psi} \in \mathbb{R}^p$, maps an input $\mathbf{x}$ to a latent representation $\mathbf{z} \in \mathcal{R} \subseteq \mathbb{R}^m$, i.e., $\mathbf{z} = f_{\boldsymbol{\psi}}(\mathbf{x})$. The linear output layer, $g_{\mathbf{W}} : \mathcal{R} \to \mathbb{R}^k$, computes the $k$ logits using parameters $\mathbf{W} \in \mathbb{R}^{k \times m}$, i.e., $g_{\mathbf{W}}(\mathbf{z}) = \mathbf{W}\mathbf{z}$. Thus, we can express the neural network $h_{\boldsymbol{\theta}} = g_{\mathbf{W}} \circ f_{\boldsymbol{\psi}}$ with parameters $\boldsymbol{\theta} = \{\boldsymbol{\psi}, \mathbf{W}\}$. Hence, the prediction of sample $i$ is $\hat{\mathbf{y}}_i = \mathbf{W} f_{\boldsymbol{\psi}}(\mathbf{x}_i)$. Lastly, parameters $\boldsymbol{\theta} = \{\boldsymbol{\psi}, \mathbf{W}\}$ are trained by minimizing the cross-entropy loss with a softmax layer, i.e., $\mathcal{L}_{ce}(\mathcal{D}; \boldsymbol{\theta}) = \frac{1}{n} \sum_{i=1}^n \ell(\mathbf{x}_i, \mathbf{y}_i, \boldsymbol{\theta})$, where $\ell(\mathbf{x}, \mathbf{y}; \boldsymbol{\theta}) = -\sum_{j=1}^k y_j \log(\sigma_j(\hat{\mathbf{y}}))$ and $\sigma_i(\mathbf{y}) = e^{\mathbf{y}_i} / \sum_j e^{\mathbf{y}_j}$ is the softmax function $\sigma : \mathbb{R}^k \to \mathbb{R}^k$.

### 3.2 Saliency-Aware Latent Decomposition

We introduce a representation learning framework that separates latent features into *salient* (i.e., task-relevant) and *non-salient* (i.e., task-irrelevant) components. The model is trained with a structured objective that integrates classification, geometric regularization, and statistical independence constraints, producing representations that improve both predictive performance and robustness. An overview of our method is shown in Figure 2 .

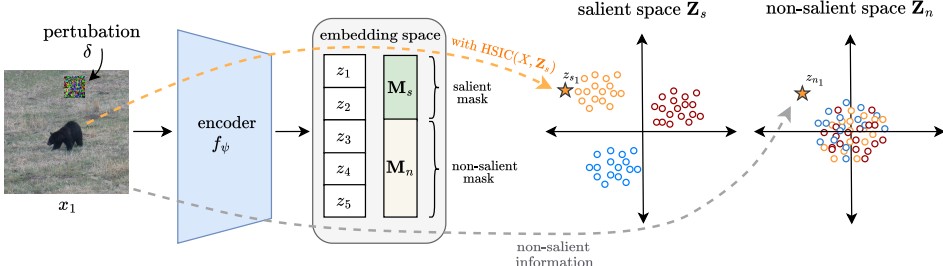

Figure 2: **Overview of H-SPLID.** The salient information for classifying the black bear is encoded in the salient space $\mathbf{z}_s$, whereas the background information is encoded in the non-salient space $\mathbf{z}_n$, allowing H-SPLID to be more robust to perturbations $\delta$ of the background.

Given the encoder $f_{\boldsymbol{\psi}}$, we introduce a learnable diagonal mask matrix $\mathbf{M}_s = \mathrm{diag}\{\boldsymbol{\beta}\} \in \{0,1\}^{m \times m}$, where $\boldsymbol{\beta} \in \{0,1\}^m$ selects salient (task-relevant) features. The complementary non-salient mask is defined as $\mathbf{M}_n = \mathbf{I} - \mathbf{M}_s$. The latent representation $\mathbf{z} = f_{\boldsymbol{\psi}}(\mathbf{x}) \in \mathcal{R}$ is then decomposed into the following salient and non-salient representations: $\mathbf{z}_s = \mathbf{M}_s\mathbf{z} = \boldsymbol{\beta} \odot \mathbf{z}$ and $\mathbf{z}_n = \mathbf{M}_n\mathbf{z} = (\mathbf{1} - \boldsymbol{\beta}) \odot \mathbf{z}$, with $\mathbf{1}$ being a vector of 1.

Define $(\mathbf{y}_i)_j$ as the $j$-th element of the $\mathbf{y}_i$ label. Then, classification is performed using *only the salient component of the latent space* $\mathbf{z}_s \in \mathcal{Z}$, and the corresponding cross-entropy loss is given by:

$$\mathcal{L}_{\mathrm{ce}}(\mathcal{D}; \boldsymbol{\theta}, \mathbf{M}_s) = -\frac{1}{n}\sum_{i=1}^{n}(\mathbf{y}_i)_j \log(\sigma(\hat{\mathbf{y}}_i)), \quad \text{with} \quad \hat{\mathbf{y}} = \mathbf{W}^{\top}\mathbf{M}_s f_{\boldsymbol{\psi}}(\mathbf{x}_i). \tag{1}$$

### 3.3 Regularizing and Preserving the Separated Features

While the saliency masks enable latent space partitioning, the quality and stability of this separation depend on additional constraints that encourage discriminative utility. We achieve this via two regularization mechanisms: masked clustering losses and Hilbert-Schmidt Independence Criterion (HSIC) [16] penalties.

Let $\mathbf{X} = [\mathbf{x}_1, \dots, \mathbf{x}_n] \in \mathbb{R}^{d \times n}$ be the matrix of input vectors and $\mathbf{Z} \equiv f_{\boldsymbol{\psi}}(\mathbf{X}) \in \mathbb{R}^{m \times n}$ be the matrix of latent vectors for $n$ samples, with masked variants $\mathbf{z}_s = \mathbf{M}_s\mathbf{Z}$ and $\mathbf{Z}_n = \mathbf{M}_n\mathbf{Z}$. For each class $k$, let $C_k$ denote the set of indices with label $k$, and define $\boldsymbol{\mu}_k$ as the class centroid and $\boldsymbol{\mu}$ as the global centroid of latent vectors. We define the following masked norm-based losses [37]:

$$\mathcal{L}_s(\mathcal{D}; \boldsymbol{\psi}, \mathbf{M}_s) = \sum_{k=1}^{K}\sum_{i \in C_k}\|\mathbf{M}_s(\mathbf{z}_i - \boldsymbol{\mu}_k)\|^2, \qquad \mathcal{L}_n(\mathcal{D}; \boldsymbol{\psi}, \mathbf{M}_n) = \sum_{i=1}^{n}\|\mathbf{M}_n(\mathbf{z}_i - \boldsymbol{\mu})\|^2. \tag{2}$$

Loss $\mathcal{L}_s$ encourages the clustering of class-specific representations in the salient subspace, strengthening its discriminative capacity. Each class has a simple uni-modal form which further removes redundant information. The loss $\mathcal{L}_n$ aligns the other features globally across samples in the non-salient space and captures shared variation. Moreover, by capturing task-irrelevant variations—such as background information—non-salient features help isolate predictive factors in the salient space, enhancing robustness and disentanglement. To further promote robust but accurate decompositions, we incorporate two additional HSIC terms:

$$\rho_s\widehat{\mathrm{HSIC}}(\mathbf{X}, \mathbf{Z}_s) + \rho_n\widehat{\mathrm{HSIC}}(\mathbf{Y}, \mathbf{Z}_n). \tag{3}$$

As HSIC is a measure of similarity (see Appendix A), the term $\widehat{\mathrm{HSIC}}(\mathbf{X}, \mathbf{Z}_s)$ reduces the dependence between the input features and the salient subspace, thereby removing redundant information. Similarly, $\widehat{\mathrm{HSIC}}(\mathbf{Y}, \mathbf{Z}_n)$ reduces the dependence between label information in the non-salient subspace. We use the unbiased empirical estimator [16] of HSIC:

$$\widehat{\mathrm{HSIC}}(\mathbf{X}, \mathbf{Z}_s) = \sum_{j=1}^{k}\sum_{i=1}^{k}(n-1)^{-2}\,\mathrm{tr}(K_x H K_z H), \tag{4}$$

where $K_x$ and $K_z$ have elements $K_{x_{(i,j)}} = k_x(\mathbf{x}_i, \mathbf{x}_j)$ and $K_{z_{(i,j)}} = k_z(\mathbf{M}_s\mathbf{z}_i, \mathbf{M}_s\mathbf{z}_j)$, and $H = \mathbf{I} - \frac{1}{n}\mathbf{1}\mathbf{1}^\top$ is the centering matrix.

Putting everything together, we define H-SPLID's overall training objective as:

$$\mathcal{L}(\mathcal{D}; \boldsymbol{\theta}, \mathbf{M}_s, \mathbf{M}_n) = \lambda_{ce}\mathcal{L}_{ce}(\mathcal{D}; \boldsymbol{\theta}, \mathbf{M}_s) + \lambda_s\mathcal{L}_s(\mathcal{D}; \boldsymbol{\psi}, \mathbf{M}_s) + \lambda_n\mathcal{L}_n(\mathcal{D}; \boldsymbol{\psi}, \mathbf{M}_n)$$
$$+ \rho_s\,\mathrm{HSIC}(\mathbf{X}, \mathbf{Z}_s) + \rho_n\,\mathrm{HSIC}(\mathbf{Y}, \mathbf{Z}_n), \tag{5}$$

where $\mathbf{Z}_s, \mathbf{Z}_n \in \mathbb{R}^{m \times n}$ are the concatenated salient and non-salient latent representation, and $\lambda_{ce}, \lambda_s, \lambda_n, \rho_s, \rho_n \geq 0$ are scalar weights. Training amounts to solving the following constrained optimization problem:

$$\min_{\boldsymbol{\theta}, \mathbf{M}_s} \quad \mathcal{L}(\mathcal{D}; \boldsymbol{\theta}, \mathbf{M}_s, \mathbf{M}_n) \tag{6a}$$

$$\text{subject to} \quad \mathbf{M}_n = I - \mathbf{M}_s, \quad \mathbf{z}_i = f_{\boldsymbol{\psi}}(\mathbf{x}_i), \ \forall i \in \{1, \ldots, n\}. \tag{6b}$$

$$\boldsymbol{\mu}_k = \frac{1}{|C_k|}\sum_{i \in C_k} \mathbf{z}_i, \quad \boldsymbol{\mu} = \frac{1}{n}\sum_{i=1}^{n} \mathbf{z}_i. \tag{6c}$$

### 3.4 The H-SPLID Algorithm

We solve Problem (6) using an alternating optimization procedure over the neural network parameters $\boldsymbol{\theta}$ and the diagonal mask matrix $\mathbf{M}_s \in \mathbb{R}^{m \times m}$ (see Algorithm 1 in Appendix B). At each outer epoch $t$, the procedure consists of two alternating steps:

**(a) Latent Representation Update (Fix $\mathbf{M}_s$, optimize $\boldsymbol{\theta}$):** Given a fixed mask $\mathbf{M}_s^{(t-1)}$, we update the encoder parameters $\boldsymbol{\theta} = \{\boldsymbol{\psi}, \mathbf{W}\}$ by minimizing the loss $\mathcal{L}$ as in Eq. (5) using minibatch stochastic gradient descent with $\mathcal{B} \subset \mathcal{D}$ for an epoch:

$$\boldsymbol{\theta}^{(t)} \leftarrow \boldsymbol{\theta}^{t-1} - \eta\nabla_{\boldsymbol{\theta}}\mathcal{L}(\mathcal{B}; \boldsymbol{\theta}^{(t-1)}, \mathbf{M}_s^{(t-1)}, \mathbf{M}_n^{(t-1)})$$

where the class means $\boldsymbol{\mu}_k$ and global means $\boldsymbol{\mu}$ are computed based on the minibatch via Eq. (6c).

**(b) Mask Update (Fix $\boldsymbol{\theta}$, optimize $\mathbf{M}_s$):** With updated latent representations $\mathbf{z}_i^{(t)} = f_{\boldsymbol{\psi}^{(t)}}(\mathbf{x}_i)$, we optimize the following optimization problem to learn the masks $\mathbf{M}_s^{(t)}, \mathbf{M}_n^{(t)}$:

$$\min_{\boldsymbol{\beta}} \quad \lambda_s\mathcal{L}_s(\mathbf{Z}^{(t)}, \mathbf{M}_s) + \lambda_n\mathcal{L}_n(\mathbf{Z}^{(t)}, \mathbf{M}_n) \tag{7a}$$

$$\text{subject to} \quad \mathbf{M}_s = \mathrm{diag}\{\boldsymbol{\beta}\}, \quad \mathbf{M}_n = \mathbf{I} - \mathbf{M}_s. \tag{7b}$$

Miklautz et al. [37] show that Prob. 7 has a closed-form solution:

$$\beta_i^* = \frac{\lambda_n\sum_{\mathbf{z}\in\mathcal{D}}(\mathbf{z}_i - \boldsymbol{\mu}_i)^2}{\lambda_s\sum_{k=1}^{K}\sum_{\mathbf{z}\in C_k}(\mathbf{z}_i - (\boldsymbol{\mu}_k)_i)^2 + \lambda_n\sum_{\mathbf{z}\in\mathcal{D}}(\mathbf{z}_i - \boldsymbol{\mu}_i)^2}, \quad \forall i \in \{1, \ldots, m\},$$

where the global mean $\boldsymbol{\mu}$ and class means $\boldsymbol{\mu}_k$ are computed from the full dataset via Eq. (6c). We use a moving average when updating the masks to improve convergence (See Algorithm 1 line 11 in Appendix B). The above optimization yields a continuous mask $\boldsymbol{\beta} \in [0, 1]^m$. After convergence, we obtain a binary version by thresholding each entry at $0.5$. In practice, using the continuous mask directly gives similar results, as the learned values typically concentrate near $0$ or $1$.

### 3.5 Theoretical Guarantees

In our experiments, we study whether H-SPLID relies on salient vs. non-salient features by examining the trained network's response to perturbations to task-irrelevant portions of the input (e.g., the right digit in Fig. 1, an image background in the COCO dataset in Section 4, etc.). Wang et al. [56] showed that HSIC regularization terms promote feature invariance and improve robustness even without adversarial training; we extend their analysis by integrating HSIC regularization (Eq. (3)) with salient space isolation (Eq. (2)), which structurally separates class-discriminative and redundant information in the representation space. To do so, we make the following assumptions on the kernel families used in the regularization terms Eq. (3):

**Assumption 3.1** (Kernel Function Boundedness and Universality). Let $K_x : \mathcal{X} \times \mathcal{X} \to \mathbb{R}^{d \times d}$ and $K_z : \mathcal{Z} \times \mathcal{Z} \to \mathbb{R}^{k \times k}$ be continuous positive-definite kernels defined on compact metric spaces $\mathcal{X}$ and $\mathcal{Z}$, respectively. Let $\mathcal{F}$ and $\mathcal{G}$ denote their associated RKHSs. We assume that:

1. The kernels are *universal* on $\mathcal{X}$ and $\mathcal{Z}$, i.e., $\mathcal{F}$ is dense in $\mathcal{C}(\mathcal{X}, \mathbb{R}^d)$ and $\mathcal{G}$ is dense in $\mathcal{C}(\mathcal{Z}, \mathbb{R}^k)$ under the supremum norm topology;

2. All functions in these RKHSs are uniformly bounded in the pointwise 2-norm, that is,
$$K_{\mathcal{F}} := \sup_{f \in \mathcal{F}} \|f\|_{\infty,2} < \infty, \quad \text{and} \quad K_{\mathcal{G}} := \sup_{g \in \mathcal{G}} \|g\|_{\infty,2} < \infty, \qquad (8)$$
where $\|f\|_{\infty,2} := \sup_{\mathbf{x} \in \mathcal{X}} \|f(\mathbf{x})\|_2$.

Many widely used kernels are known to be universal on compact subsets of $\mathbb{R}^d$, including the Gaussian (RBF), Laplacian, and Matérn kernels [46, 45]. Universality ensures that the RKHS is rich enough to approximate any continuous function on the domain, while boundedness holds automatically on compact input spaces when the kernel is continuous. These properties collectively justify the use of kernel-based function classes for comparing or aligning with neural network outputs, particularly when both are assumed to operate over compact, bounded input spaces.

As is common when modeling bounded inputs [33, 40], we use the truncated multivariate normal (tMVN) distribution: $\mathbf{x} \sim \mathcal{N}_R(0, \sigma^2 I_d)$, with density $\tilde{p}(\mathbf{x}) = \frac{1}{C} \exp\left(-\frac{\|\mathbf{x}\|^2}{2\sigma^2}\right) \cdot \mathbf{1}_{\|\mathbf{x}\| \leq R}$, where $R > 0$ denotes the truncation radius, $\sigma^2 > 0$ is the variance parameter, and $C$ is the normalization constant. Our main robustness guarantee shows that the sensitivity of the model prediction is controlled by (a) the dimensionality of the salient space and (b) the HSIC between the inputs and salient representations.

**Theorem 3.2** (HSIC-Based Robustness Bound). *Let $\mathbf{x}$ be sampled from a tMVN distribution $\mathbf{x} \sim \mathcal{N}_R(0, \sigma^2 I_d)$, and let the neural network $h_{\boldsymbol{\theta}} : \mathbb{R}^d \to \mathbb{R}^k$ be differentiable almost everywhere with an L-Lipschitz encoder $f_{\boldsymbol{\psi}}$ and a bounded linear output layer with $\|\mathbf{W}\|_{\infty} \leq B$. Suppose the RKHSs associated with $K_x$ and $K_z$ satisfy Assumption 3.1, with kernel sup-norm bounds $K_{\mathcal{F}}, K_{\mathcal{G}}$. Let $s := \|\mathbf{M}_s\|_0$ be the count of non-zero entries of the salient mask. Then, for all perturbation maps $\delta : \mathbb{R}^d \to \mathbb{R}^d$ such that $\|\delta(\mathbf{x})\|_2 \leq r$ for all $\mathbf{x} \in \mathcal{X}$, the expected output deviation satisfies*

$$\mathbb{E}_{\mathbf{x}}\left[\|h_{\boldsymbol{\theta}}(\mathbf{x} + \delta(\mathbf{x})) - h_{\boldsymbol{\theta}}(\mathbf{x})\|_2\right] \leq \frac{rRB\sqrt{ks}(LR + \|f_{\boldsymbol{\psi}}(0)\|_2)}{\sigma^2 K_{\mathcal{F}} K_{\mathcal{G}}} \cdot \mathrm{HSIC}(\mathbf{x}, \mathbf{z}_s) + o(r), \quad (9)$$

*where $\mathbf{z}_s := \mathbf{M}_s f_{\boldsymbol{\psi}}(\mathbf{x})$ is the salient representation and $\boldsymbol{\theta} = \{\mathbf{W}, \boldsymbol{\psi}\}$ denotes the collection of neural network parameters.*

The proof of Theorem 3.2 can be found in Appendix C. Intuitively, with stronger information compression imposed by minimizing both the HSIC term (thereby reducing $\mathrm{HSIC}(\mathbf{x}, \mathbf{z}_s)$) and the masks (thereby reducing the salient mask support $s := \|\mathbf{M}_s\|_0$), the model is forced to rely only on salient features: Theorem 3.2 suggests that, in this case, the perturbation is more likely to end up in the non-salient space, and the majority of the attack does not contribute to the change of the output of the neural network, as $\|h_{\boldsymbol{\theta}}(\mathbf{x} + \delta(\mathbf{x})) - h_{\boldsymbol{\theta}}(\mathbf{x})\|$ stays small for any perturbation map $\delta(\cdot)$ with $\|\delta(\mathbf{x})\|_2 \leq r$. From a technical perspective, our theorem differs from Wang et al. [56] in two aspects. First, we sharpen the dependence of the upper bound on the power of the perturbation; this allows us to explicitly link it to the dimension of the salient mask $s$. Second, we extend their binary classification framework (i.e., $k = 1$ with one output value) to multi-class classification (with arbitrary $k$) to cover a wider range of classification models.

Additionally, we can quantify how perturbations will trigger prediction changes from the volume of the entire input domain. In particular, we define the salient-active region as $\mathcal{X}_s(\epsilon) := \{\mathbf{x} \in \mathcal{X}_R : \|\nabla_{\mathbf{x}} h_{\boldsymbol{\theta}}(\mathbf{x})\|_F > \epsilon\}$. Under the tMVN distribution, the probability that an input falls into this region equals its measure: $\mu(\mathcal{X}_s(\epsilon)) = \mathbb{P}(\|\nabla_{\mathbf{x}} h_{\boldsymbol{\theta}}(\mathbf{x})\|_F > \epsilon)$. We can bound this probability as follows:

**Corollary 3.3** (Salient Region Volume Bound via HSIC). *Under the same assumptions of Theorem 3.2, for any threshold $\epsilon > 0$, the probability that a random input falls into the salient-active input region is upper bounded by*

$$\mathbb{P}_{\mathbf{x}}(\|\nabla_{\mathbf{x}} h_{\boldsymbol{\theta}}(\mathbf{x})\|_F > \epsilon) \leq \frac{1}{\epsilon} \cdot \left(\frac{RB\sqrt{ks}(LR + \|f_{\boldsymbol{\psi}}(0)\|_2)}{\sigma^2 K_{\mathcal{F}} K_{\mathcal{G}}} \cdot \mathrm{HSIC}(\mathbf{x}, \mathbf{z}_s) + o(1)\right). \quad (10)$$

| Dataset | Encoder Model $f_\psi$ | Input size | Perturbation type | # Categories |
|---|---|---|---|---|
| C-MNIST | LeNet-3 | $1 \times 64 \times 64$ | PGD attack on right digit [34] | 10 |
| COCO subset (bear, elephant, giraffe, zebra) | ResNet-18 | $3 \times 224 \times 224$ | PGD and AA attack (block, background, full) [10] | 4 |
| ISIC-2017 (nevus, melanoma, seborrheic keratosis) | ResNet-50 | $3 \times 224 \times 224$ | Real-world corruptions (brightness, defocus, occlusion) [21] | 3 |
| ImageNet-9 | ResNet-50 | $3 \times 224 \times 224$ | Background manipulation and removal | 368 |
| CounterAnimal (Common vs. Counter) | ResNet-50 | $3 \times 224 \times 224$ | Counterfactual backgrounds | 45 |

Table 1: **Datasets and corresponding models**. "Perturbation type" summarizes how backgrounds/contexts are manipulated in our evaluations.

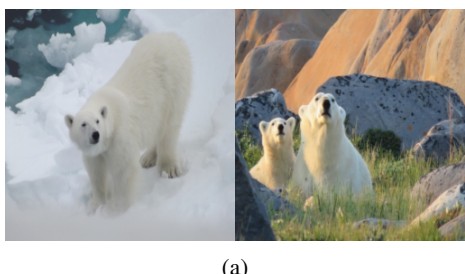
(a)

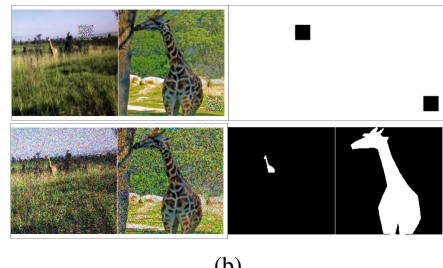
(b)

Figure 3: (a) **Samples from the CounterAnimal dataset.** Common set (left), counter set (right). (b) **Adversarial attacks on non-salient features.** We attack the non-salient background of COCO images (left) given their corresponding block (upper right) or background mask (lower right) to test whether models successfully learned salient features.

Thus, Corollary 3.3 implies that the volume of the salient-active region is tightly controlled by the dimensionality of the salient space and the HSIC between inputs and the salient representation. The proof is in Appendix E.

## 4 Experiments

In this section, we describe our datasets, comparison methods, experiment setup, and performance metrics. Additional runtime experiments, implementation details, and hyperparameter tuning and configuration protocols provided in the Appendix F. Our code is publicly available at `https://github.com/neu-spiral/H-SPLID`.

**Datasets and Encoder Models**    We evaluate H-SPLID on synthetic and natural image benchmarks, spanning five datasets and three architectures (Table 1). We create a synthetic Concatenated-MNIST (C-MNIST) dataset (see Fig. 1) by concatenating two MNIST digits with the left digit as the class label. We use LeNet-3 [27, 26] as an encoder. We construct a four-class subset of COCO [28] (*bear*, *elephant*, *giraffe*, *zebra*), coupled with a ResNet-18 [20] encoder. ISIC-2017 is a medical imaging dataset [9]. ImageNet-9 (IN-9) [58], encompasses 368 classes from ImageNet-1K instantiated in three variants: *Original* images, a *MixedRand* variant in which object foregrounds are put onto random-class backgrounds, and an *Only-FG* variant with backgrounds entirely removed (See Figure 4). CounterAnimal (CA) [55], splits iNaturalist wildlife photos into a Common set (exhibiting typical backgrounds) and a Counter set (featuring atypical yet plausible backgrounds) (see Fig. 3a). We use ResNet-50 as an encoder for ISIC-2017 and ImageNet derived datasets.

**Comparison Methods.**    To demonstrate how H-SPLID focuses on task-relevant characteristics, we compare it with several methods for feature selection and weight regularization, including *weight decay* ($L_2$ regularization) [25], $L_1$ regularization [49], *Group-Lasso* regularization [62], and two activation-sparsity variants – one applying $L_1$ penalty to the penultimate layer's activations and another applying a *Group-Lasso* penalty across those activations to promote instance-level sparsity. Finally, we compare against HBaR [56] under non-adversarial training, and a *vanilla* baseline trained only with cross-entropy loss. Appendix F provides further details on training and implementation.

---

[1]CounterAnimal is a benchmark split (Common/Counter) over multiple species; we follow its predefined taxonomy and report performance across the two splits rather than a fixed class count.

| | No Atk. | Block Atk. | | | Background Atk. | | | Full Atk. | |
|---|---|---|---|---|---|---|---|---|---|
| Comp. | $\epsilon=0$ | $\frac{25}{255}$ | $\frac{128}{255}$ | $\frac{255}{255}$ | $\frac{1}{255}$ | $\frac{2}{255}$ | $\frac{3}{255}$ | $\frac{1}{255}$ | $\frac{2}{255}$ |
| **PGD** Va. | **98.1±0.4** | 56.3±0.6 | 51.3±1.3 | 55.1±0.5 | 75.2±0.2 | 56.6±0.2 | 34.4±0.1 | 55.9±0.2 | 34.2±0.3 |
| WD | 94.3±0.7 | 43.9±0.5 | 57.2±0.4 | 76.9±0.8 | 76.3±0.0 | 59.9±0.3 | 43.0±0.1 | 57.8±0.1 | 40.7±0.2 |
| GLA | 97.1±0.6 | 60.4±1.2 | **70.4±0.8** | **78.8±0.8** | 75.1±0.2 | 57.4±0.3 | 35.3±0.3 | 57.5±0.3 | 37.3±0.3 |
| GLW | 92.6±1.1 | 45.4±0.5 | 47.2±0.5 | 58.0±0.6 | 72.6±0.0 | 58.4±0.0 | 42.9±0.1 | 54.9±0.1 | 41.2±0.2 |
| LSA | 97.1±0.4 | 57.3±1.2 | 63.5±0.5 | 71.9±0.6 | 71.2±0.1 | 54.1±0.3 | 33.7±0.2 | 51.7±0.2 | 35.0±0.4 |
| LSW | 96.0±0.6 | 43.2±0.5 | 42.5±0.8 | 57.0±0.4 | 73.5±0.0 | 55.6±0.1 | 37.0±0.1 | 53.1±0.1 | 33.5±0.1 |
| HBaR | 97.1±0.4 | 57.4±1.3 | 54.0±1.1 | 67.0±1.2 | 77.9±0.1 | 60.2±0.3 | 39.9±0.3 | 62.4±0.2 | 41.9±0.3 |
| Ours | 97.9±0.3 | **71.9±0.7** | 68.5±0.5 | 72.3±0.4 | **78.0±0.1** | **68.9±0.5** | **57.5±0.2** | **66.5±0.1** | **58.9±0.4** |
| **AA** Va. | **98.1±0.4** | 42.8±0.2 | 21.0±0.8 | 19.9±0.6 | 66.8±0.1 | 41.6±0.1 | 26.4±0.2 | 45.5±0.1 | 20.9±0.1 |
| WD | 94.3±0.7 | 38.5±0.8 | 23.0±0.7 | 22.5±0.6 | 74.1±0.0 | 53.4±0.1 | 38.7±0.1 | 53.5±0.0 | 29.3±0.0 |
| GLA | 97.1±0.6 | 43.3±1.0 | 27.2±1.0 | 28.9±1.2 | 64.6±0.1 | 39.8±0.1 | 26.8±0.2 | 45.5±0.0 | 21.0±0.1 |
| GLW | 92.6±1.1 | 41.3±0.8 | 24.6±0.8 | 23.1±0.6 | 71.1±0.1 | 52.0±0.0 | 38.8±0.1 | 52.1±0.0 | 29.5±0.0 |
| LSA | 97.1±0.4 | 40.9±0.7 | 25.5±0.6 | 25.7±0.6 | 62.6±0.1 | 39.5±0.1 | 25.3±0.2 | 42.4±0.1 | 20.3±0.1 |
| LSW | 96.0±0.6 | 37.5±0.6 | 20.6±0.7 | 20.5±0.5 | 69.7±0.0 | 47.3±0.1 | 31.8±0.2 | 47.3±0.0 | 20.5±0.1 |
| HBaR | 97.1±0.4 | 39.5±0.9 | 21.4±0.8 | 29.2±0.4 | 70.1±0.1 | 45.4±0.1 | 31.4±0.3 | 50.4±0.1 | 25.2±0.2 |
| Ours | 97.9±0.3 | **62.1±0.4** | **48.8±0.3** | **48.3±0.5** | **74.2±0.0** | **59.6±0.2** | **52.6±0.1** | **60.4±0.1** | **48.8±0.2** |

Table 2: **Measuring saliency with adversarial attacks on COCO**. H-SPLID (Ours) improves robustness to adversarial attacks compared to most baselines, with the largest performance gains observed under stronger background-targeted attacks (two middle columns). Here, AA denotes AutoAttack, while PGD denotes Projected Gradient Descent. The attack magnitude $\epsilon$ is indicated using ratios of pixel values, with the strongest attack being $\frac{255}{255}$. All models are trained without adversarial data. Va., WD, LSA, LSW, GLA, and GLW denote Vanilla, Weight Decay, $L_1$ Sparse Activations, $L_1$ Sparse Weights, Group-Lasso Activations, and Group-Lasso Weights, respectively.

We train H-SPLID and all comparison methods exclusively on clean data without employing adversarial attacks or having access to saliency masks. In all datasets, we employ a 80-20 validation split for tuning, and use held-out test sets for final evaluation. Following prior art [32, 60, 59, 56], we use the Normalized Cross Covariance Operator [14] to get a scale-insensitive HSIC penalty. All methods are evaluated using clean test accuracy (over three seeds) and robust test accuracy under different attacks, described below.

**Testset Attacks.** Methods are evaluated w.r.t. a broad array of attacks on non-salient features at test-time. On C-MNIST, we evaluate predictive performance against a PGD attack on the (non-salient) right digit. For COCO experiments, we pretrain a ResNet-18 [20] from random initialization for 100 epochs with cross-entropy, followed by 200 epochs of method-specific training before evaluating on the held-out test set. We test PGD and AA in three ways: random blocks of pixels in the background, pixel perturbations in the background, and full-image attacks. On ISIC-2017, we use a ResNet-50 pretrained on ImageNet for feature extraction, train a three-class head for 50 epochs, and then run 50 epochs of method-specific training. We test robustness under real-world corruptions (brightness, defocus blur, and snow/occlusion from the corruptions benchmark [21]) applied to non-salient regions (non-lesion pixels). We use IN-9 and CA for transfer learning experiments as follows. First, we train a ResNet-50 [20] initialized from ImageNet-1K pretrained weights (TorchVision [35]) for 20 epochs of method-specific training on ImageNet-1K. Then, we test the model on the IN-9 (the original IN-9 and its MixedRand and Only-FG variants) and also on CA (CA-Common and CA-Counter) evaluation sets (see Table 4).

PGD is implemented via 10 iterations with a step size $\alpha = 0.0156$ and AutoAttack [10] is implemented using the rand ensemble. Attacks per block (Block Atk., see Fig. 3b) are confined to a single randomly placed block in the background, with size $\frac{1}{4} \times \frac{1}{4}$ of the image dimensions. Attacks restricted to background pixels (Block Atk., Background Atk., see Fig. 3b) use saliency masks, which are available for COCO and ISIC-2017. Full attacks (Full Atk.) are across the entire image. PGD and AA Attacks are conducted over a range of $\epsilon$ levels, with each configuration repeated across five random seeds. Additional implementation details and hyperparameter settings are provided in Appendix F.

|      | No Perturb. | Brightness | Defocus | Occlusion |
|------|-------------|------------|---------|-----------|
| Va.  | 75.45±0.986 | 66.43±2.527 | 63.77±2.388 | 62.87±3.081 |
| WD   | 75.63±1.545 | 67.53±2.980 | 64.57±2.295 | 63.55±3.851 |
| LSA  | 75.62±1.211 | 66.50±2.171 | 64.33±1.653 | 62.27±4.256 |
| LSW  | 75.30±1.040 | 66.13±2.432 | 63.22±2.506 | 62.70±3.810 |
| GLA  | 75.38±1.383 | 66.50±3.136 | 61.32±4.501 | 62.68±2.969 |
| GLW  | 70.65±4.118 | 60.23±6.698 | 58.63±9.564 | 60.62±5.697 |
| HBaR | 75.90±0.844 | 68.70±1.942 | 65.62±2.058 | 66.18±3.013 |
| Ours | **76.78±0.778** | **70.00±1.619** | **68.38±1.376** | **69.50±1.716** |

Table 3: **Measuring saliency with real-world perturbations on ISIC-2017**. H-SPLID (Ours) achieves the best robustness across lighting (brightness), blur (defocus), and occlusion (snow) when perturbations are restricted to non-salient regions. All models are trained without adversarial data. Va., WD, LSA, LSW, GLA, and GLW denote Vanilla, Weight Decay, $L_1$ Sparse Activations, $L_1$ Sparse Weights, Group-Lasso Activations, and Group-Lasso Weights, respectively.

## 4.1 Results

**Controlled attack Benchmark COCO.** We quantitatively demonstrate the ability of H-SPLID to learn salient features on the four-class COCO benchmark by evaluating adversarial robustness under block, background, and full-image perturbations. As shown in Table 2, H-SPLID achieves 57.5% under background-only PGD attacks at $\epsilon = 3/255$, with the closest competitor attaining 43.0%. Even when attacks span the entire image, H-SPLID sustains 58.9% accuracy under a PGD attack with $\epsilon = 2/255$, surpassing the 34.2% of the vanilla network and 41.9% of the best performing competitor. Against the stronger AutoAttack ensemble, H-SPLID consistently outperforms all baselines in robustness to adversarial perturbations.

These results show that explicitly decomposing latent features into salient and non-salient subspaces delivers substantial robustness gains, with the most pronounced improvements occurring under background-only perturbations, validating that H-SPLID effectively isolates redundant information. Moreover, robustness gains are achieved without any adversarial training, demonstrating that H-SPLID's latent decomposition strategy yields inherently saliency preserving representations.

**Medical imaging Benchmark ISIC-2017.** To further assess domain generality and robustness, we evaluate H-SPLID on the ISIC-2017 skin lesion classification dataset [9] (three classes: nevus, melanoma, seborrheic keratosis). We perturb only non-salient regions (e.g., non-lesion pixels) and adopt three real-world corruptions from the corruptions benchmark [21]: *brightness* (lighting), *defocus blur* (blur), and *snow* (which effectively occludes small patches; we report it as *occlusion*). Results are averaged over 10 random seeds.

These medical imaging results mirror our COCO findings: explicitly separating salient from non-salient latents confers consistent robustness gains under realistic, non-adversarial corruptions, especially when perturbations target only non-salient regions. Together with Table 2, this strengthens the evidence that H-SPLID learns saliency-preserving representations that generalize beyond natural images to specialized clinical domains.

**Saliency Benchmarks.** We measure the saliency of our model on the ImageNet-9 and CounterAnimal benchmarks. In Table 4, H-SPLID attains the highest accuracy on the IN-9 test set (76.7%), outperforming the vanilla baseline by 2.7% and exceeding the next best regularization method by over 1%. When the backgrounds are entirely removed (Only-FG), H-SPLID once again surpasses all methods with a 64.5% test accuracy, demonstrating its ability to distill object-centric features. On the more challenging MixedRand variant, it achieves a 59.5% test accuracy, a substantial 3.1% gain over the strongest baseline. On the CA Common set, which preserves typical contextual correlations, H-SPLID matches the top performing method (80.3% vs. 80.7%). Finally, on the CA Counter set of atypical contexts, it surpasses all competitors with a 60.6% test accuracy, a 2.1% improvement over the HBaR model. The consistent performance across original, background-altered and contextually shifted datasets demonstrates that the explicit separation of salient and non-salient subspaces in H-SPLID yields representations that transfer more robustly to new tasks and real-world perturbations.

| Method | IN-9 | Only-FG | MixedRand | CA-Common | CA-Counter |
|---|---|---|---|---|---|
| Vanilla | 74.0 | 60.5 | 51.2 | 78.3 | 58.4 |
| Weight Decay | 72.6 | 58.4 | 51.2 | 77.3 | 54.4 |
| Group Lasso Activations | 75.3 | 63.8 | 55.7 | 79.9 | 58.3 |
| Group Lasso Weights | 73.0 | 60.0 | 50.6 | 78.3 | 57.1 |
| $L_1$ Sparse Activations | 74.8 | 62.9 | 56.4 | **80.7** | 58.1 |
| $L_1$ Sparse Weights | 73.7 | 61.3 | 51.9 | 78.1 | 54.7 |
| HBaR | 73.6 | 63.3 | 53.8 | 79.3 | 58.5 |
| H-SPLID | **76.7** | **64.5** | **59.5** | 80.3 | **60.6** |

Table 4: **Transfer accuracy on ImageNet-9 and CounterAnimal saliency benchmarks** H-SPLID achieves the highest accuracy on the most challenging splits (MixedRand and CA-Counter), demonstrating the robust transferability of its learned representations.

| Method | No Atk. | Background Atk. | Full Atk. |
|---|---|---|---|
| $\lambda_{ce}\mathcal{L}_{ce}$ | **98.30** | 33.75±0.1 | 35.29±0.5 |
| $\lambda_{ce}\mathcal{L}_{ce} + \lambda_s\mathcal{L}_s + \lambda_n\mathcal{L}_n$ | 97.52 | 43.69±0.2 | 44.12±0.3 |
| $\lambda_{ce}\mathcal{L}_{ce} + \rho_s\text{HSIC}(\mathbf{X}, \mathbf{Z}_s) + \rho_n\text{HSIC}(\mathbf{Y}, \mathbf{Z}_n)$ | 96.74 | 42.71±0.3 | 45.87±0.8 |
| H-SPLID (full $\mathcal{L}$) | 97.59 | **57.12±0.3** | **58.44±0.2** |

Table 5: **Ablation of loss terms on COCO.** Accuracy under no attack, Background Attack ($\epsilon = 3/255$), and Full Attack ($\epsilon = 2/255$) using PGD. Attacks run with five random seeds. The complete objective delivers the highest robust performance.

**Loss Term Ablations.** We ablate the loss components according to their conceptual grouping, namely cross-entropy ($\mathcal{L}_{ce}$), cross-entropy loss with space separation ($\mathcal{L}_{ce} + \mathcal{L}_s + \mathcal{L}_n$), cross-entropy loss with HSIC components ($\mathcal{L}_{ce} + \text{HSIC}(\mathbf{X}, \mathbf{Z}_s) + \text{HSIC}(\mathbf{Y}, \mathbf{Z}_n)$) and the full H-SPLID loss ($\mathcal{L}$). The mask computation (Section 3.4) remains unchanged across all ablations (the difference is whether the clustering loss terms $\mathcal{L}_s + \mathcal{L}_n$ are optimized). All ablations are performed starting from the best-performing COCO model (Appendix F.4) by removing individual components of the full $\mathcal{L}$ objective. Each loss combination was independently tuned to achieve its best performance. As shown in Table 5, simply using the cross-entropy loss yields poor background robustness (33.75%). Adding the $\mathcal{L}_s$ and $\mathcal{L}_n$ terms or the HSIC penalties improves robustness to approximately 43-46%, while maintaining clean accuracy above 96%. The complete objective results in the best robust performance (57.12%) while maintaining competitive clean accuracy (97.59%). We further assess the sensitivity of H-SPLID to its hyperparameters in Appendix G.2.

# 5 Limitations & Conclusion

**Limitations.** H-SPLID assumes the presence of irrelevant information in the input, as well as a sufficiently diverse dataset in which class-specific features occur across varying contexts. If a particular feature always co-occurs with the same context, H-SPLID cannot separate salient from non-salient information, since both appear inseparably – a challenge that would require external knowledge to resolve. Further, we restricted our analysis to image data, where the distinction between salient and non-salient regions is intuitive to humans. Investigating whether similar decompositions apply to other data modalities remains an exciting direction for future work.

**Conclusion.** We introduce H-SPLID, a novel method for salient feature learning that decomposes the latent space of a neural network into task-relevant and task-irrelevant components during training. Unlike prior work, H-SPLID performs supervised feature selection in an end-to-end manner, without relying on external saliency annotations. Our theoretical analysis provides formal insight into how this decomposition promotes compact and informative representations. Empirically, we show that H-SPLID learns class-discriminative features and naturally reduces reliance on irrelevant input variations. In future work, we would like to combine H-SPLID with self-supervised models such as I-JEPA [4], with the goal of learning features that generalize better to downstream tasks. Additionally, we plan to explore the decomposition of salient and non-salient spaces in other data modalities, including graphs, text, and multi-modal data.

## Acknowledgments and Disclosure of Funding

We gratefully acknowledge support from the National Science Foundation through grant CNS-2414652. Further, this work is supported in part by NIH 5U24CA264369-03. We acknowledge the EuroHPC Joint Undertaking for awarding this project access to the MareNostrum supercomputer (hosted at BSC, Spain), MeluXina (operated by LuxProvide, Luxembourg), Deucalion (hosted at the Minho Advanced Computing Center, Portugal), and Discoverer (hosted at Sofia Tech, Bulgaria) through EuroHPC Access allocations.

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

## A   Technical Preliminary

**Hilbert Space.** A Hilbert space is a complete inner product space. More formally, a real or complex vector space $\mathcal{H}$ is called a Hilbert space if it is equipped with an inner product $\langle \cdot, \cdot \rangle_{\mathcal{H}}$ that induces a norm $\|f\|_{\mathcal{H}} := \sqrt{\langle f, f \rangle_{\mathcal{H}}}$, under which $\mathcal{H}$ is complete; that is, every Cauchy sequence in $\mathcal{H}$ converges to a limit in $\mathcal{H}$. The inner product structure generalizes the geometric notions of angle and length, while completeness ensures that limits of convergent sequences remain in the space. Examples include $\mathbb{R}^n$, $L^2$ spaces of square-integrable functions, and reproducing kernel Hilbert spaces (RKHSs).

**Reproducing Kernel Hilbert Space (RKHS).** Let $\mathcal{X}$ be a nonempty set. A Hilbert space $\mathcal{H} \subseteq \mathbb{R}^{\mathcal{X}}$ is called a *reproducing kernel Hilbert space* if there exists a positive definite kernel $k : \mathcal{X} \times \mathcal{X} \to \mathbb{R}$ such that, for every $\mathbf{x} \in \mathcal{X}$, the function $k(x, \cdot) \in \mathcal{H}$ and the reproducing property holds: that is, for all $f \in \mathcal{H}$ and $x \in \mathcal{X}$,

$$f(\mathbf{x}) = \langle f, k(\mathbf{x}, \cdot) \rangle_{\mathcal{H}}.$$

**Hilbert-Schmidt Operator [16].** Let $\mathcal{F}$ and $\mathcal{G}$ be separable Hilbert spaces, and let $A : \mathcal{F} \to \mathcal{G}$ be a bounded linear operator.[2] Then $A$ is called a Hilbert-Schmidt operator if, for any orthonormal bases $\{f_i\}_{i=1}^{\infty} \subset \mathcal{F}$ and $\{g_j\}_{j=1}^{\infty} \subset \mathcal{G}$, the following Hilbert-Schmidt norm is finite:

$$\|A\|_{\mathrm{HS}}^2 := \sum_{i=1}^{\infty} \sum_{j=1}^{\infty} \langle A f_i, g_j \rangle_{\mathcal{G}}^2 < \infty. \tag{11}$$

**Cross-Covariance Operator.** Let $\mathbf{x} \in \mathcal{X}$, $\mathbf{z} \in \mathcal{Z}$ be random variables and let $\mathcal{F}$ and $\mathcal{G}$ be RKHSs over $\mathcal{X}$ and $\mathcal{Z}$. Then, the *cross-covariance operator* $C_{XZ} : \mathcal{G} \to \mathcal{F}$ is the unique linear operator such that

$$\langle f, C_{XZ} g \rangle_{\mathcal{F}} := \mathrm{Cov}[f(\mathbf{x}), g(\mathbf{z})] = \mathbb{E}\left[ (f(\mathbf{x}) - \mathbb{E}[f(\mathbf{x})])(g(\mathbf{z}) - \mathbb{E}[g(\mathbf{z})]) \right], \tag{12}$$

for all $f \in \mathcal{F}$, $g \in \mathcal{G}$.

**Proposition A.1** (Covariance Bounded by HSIC [16, 15])**.** *Let $X \in \mathcal{X}$ and $\mathbf{z} \in \mathcal{Z}$ be random variables, and let $\mathcal{F}$ and $\mathcal{G}$ be RKHSs on $\mathcal{X}$ and $\mathcal{Z}$, respectively. Then the scalar covariance is bounded by the Hilbert-Schmidt Information Criterion, i.e., the Hilbert-Schmidt norms of the cross-covariance operators:*

$$\sup_{f \in \mathcal{F}_g \in \mathcal{G}} \mathrm{Cov}[f(\mathbf{x}), g(\mathbf{z})] \leq \mathrm{HSIC}_s(\mathbf{x}, \mathbf{z}) \equiv \|C_{XZ}\|_{\mathrm{HS}}. \tag{13}$$

The above proposition is the HSIC for the scalar value RKHS defined in Gretton et al. [16]. To connect the above scalar value function spaces to vector-value spaces, we use the external direct sum as below.

**External Direct Sum of Hilbert Spaces.** Let $\mathcal{H}_1, \ldots, \mathcal{H}_k$ be Hilbert spaces. We can then denote the vector-valued Hilbert space $\mathcal{H}$ via the external direct sum as

$$\mathcal{H} := \bigoplus_{j=1}^{k} \mathcal{H}_j := \{(f_1, \ldots, f_k) \mid f_j \in \mathcal{H}_j\}, \tag{14}$$

which is equipped with the inner product $\langle (f_1, \ldots, f_k), (g_1, \ldots, g_k) \rangle_{\mathcal{H}} := \sum_{j=1}^{k} \langle f_j, g_j \rangle_{\mathcal{H}_j}$. The corresponding norm is given by $\|f\|_{\mathcal{H}} := \left( \sum_{j=1}^{k} \|f_j\|_{\mathcal{H}_j}^2 \right)^{1/2}$, which makes $\mathcal{H}$ itself a Hilbert space.

Moreover, if we construct the RKHS by the direct sum, i.e., $\mathcal{H} := \bigoplus_{j=1}^{k} \mathcal{H}_j$, the resulting space $\mathcal{H}$ is a vector-valued RKHS [7].

Next, we define the corresponding covariance matrix for a vector-valued RKHS. Let $f : \mathcal{X} \to \mathbb{R}^k$ and $g : \mathcal{Z} \to \mathbb{R}^k$ be vector-valued functions, and $(\mathbf{x}, \mathbf{z})$ be random variables jointly distributed over $\mathcal{X} \times \mathcal{Z}$. The covariance matrix between $f$ and $g$ is defined as:

$$\mathrm{Cov}[f(\mathbf{x}), g(\mathbf{z})] := \mathbb{E}\left[ (f(\mathbf{x}) - \mathbb{E}[f(\mathbf{x})])(g(\mathbf{z}) - \mathbb{E}[g(\mathbf{z})])^{\top} \right] \in \mathbb{R}^{k \times k}. \tag{15}$$

---

[2]Separable Hilbert spaces implies the spaces have a complete orthonormal basis.

---

**Algorithm 1** Alternating Optimization of $\boldsymbol{\theta}$ and $\mathbf{M}_s$

---

1: **Input:** Dataset $\mathcal{D} = \{(\mathbf{x}_i, y_i)\}_{i=1}^n$; initial $\boldsymbol{\theta}^{(0)}$; $\mathbf{M}_s^{(0)} = I$; $\beta_{step} \in [0, 1]$;
2: **for** epoch $t = 1$ to $T$ **do**
    **Step 1: Update $\theta$ via SGD (minibatches)**
3:     **for** each minibatch $\mathcal{B} \subset \mathcal{D}$ **do**
4:         Compute latent codes $\mathbf{z}_i = f_{\boldsymbol{\psi}}(\mathbf{x}_i)$ for $i \in \mathcal{B}$
5:         Compute minibatch means: $\boldsymbol{\mu}, \{\boldsymbol{\mu}_k\}_{k=1}^K$
6:         Compute loss $\mathcal{L}(\mathcal{B}; \boldsymbol{\theta}, \mathbf{M}_s, I - \mathbf{M}_s)$
7:         Update $\boldsymbol{\theta} \leftarrow \boldsymbol{\theta} - \eta \nabla_{\boldsymbol{\theta}} \mathcal{L}$
8:     **end for**
    **Step 2: Update $\mathbf{M}_s$ via closed-form solution**
9:     Compute $\mathbf{z}_i = f_{\boldsymbol{\psi}}(\mathbf{x}_i)$ for all $i \in \mathcal{D}$
10:     Compute dataset means: $\boldsymbol{\mu}, \{\boldsymbol{\mu}_k\}_{k=1}^K$
11:     **for** each feature dimension $i = 1$ to $m$ **do**

$$(\mathbf{M}_s)_{i,i} \leftarrow \beta_{step}(\mathbf{M}_s)_{i,i} + (1 - \beta_{step}) \frac{\lambda_n \sum_{\mathbf{z} \in \mathcal{D}} (\mathbf{z}_i - \boldsymbol{\mu}_i)^2}{\lambda_s \sum_{k=1}^K \sum_{\mathbf{z} \in C_k} (\mathbf{z}_i - (\boldsymbol{\mu}_k)_i)^2 + \lambda_n \sum_{\mathbf{z} \in \mathcal{D}} (\mathbf{z}_i - \boldsymbol{\mu}_i)^2}$$

12:     **end for**
13: **end for**
14: **Return:** $\boldsymbol{\theta}^{(T)}, \mathbf{M}_s^{(T)}$

---

**Hilbert-Schmidt Independence Criterion (HSIC)** The HSIC [16] is a kernel-based measure of dependence. Let $\mathbf{x} \in \mathcal{X} \subseteq \mathbb{R}^d$ and $\mathbf{z} \in \mathcal{Z} \subseteq \mathbb{R}^m$ be random variables. Let also $\mathcal{G} = \bigoplus_{i=1}^k \mathcal{G}_i$ and $\mathcal{F} = \bigoplus_{j=1}^k \mathcal{F}_j$ be vector-valued RKHSs over the input and representation domains with $k$ values, respectively (i.e., direct sums of $k$ scalar RKHSs). The *cross-covariance operator* $C_{XZ} : \mathcal{G} \to \mathcal{F}$ is the unique linear operator such that $\langle f, C_{XZ}g \rangle_{\mathcal{F}} := \mathrm{Cov}[f(\mathbf{x}), g(\mathbf{z})] = \mathbb{E}\left[(f(\mathbf{x}) - \mathbb{E}[f(\mathbf{x})])(g(\mathbf{z}) - \mathbb{E}[g(\mathbf{z})])\right]$, for all $f \in \mathcal{F}, g \in \mathcal{G}$. The vector-valued HSIC between $\mathbf{x}$ and $\mathbf{z}$ is defined as

$$\mathrm{HSIC}(\mathbf{x}, \mathbf{z}) := \sum_{j=1}^k \sum_{i=1}^k \|C_{XZ}^{(i,j)}\|_{\mathrm{HS}}, \tag{16}$$

where $C_{XZ}^{(i,j)}$ is the cross-covariance operator between $\mathcal{G}_i$ and $\mathcal{F}_j$, and $\| \cdot \|_{\mathrm{HS}}$ denotes the Hilbert-Schmidt norm. This quantity upper-bounds the scalar covariances [15]:

$$\sup_{f_j \in \mathcal{F}_j, \, g_i \in \mathcal{G}_i} \mathrm{Cov}[f_j(\mathbf{x}), g_i(\mathbf{z})] \leq \|C_{XZ}^{(i,j)}\|_{\mathrm{HS}}.$$

We can empirically estimate HSIC within $O(n^{-1})$ accuracy [16] given $n$ i.i.d. samples $\{(x_i, y_i)\}_{i=1}^n$ via:

$$\widehat{\mathrm{HSIC}}(\mathbf{X}, \mathbf{Z}) = \sum_{j=1}^k \sum_{i=1}^k (n-1)^{-2} \mathrm{tr}(K_x H K_z H), \tag{17}$$

where $K_x$ and $K_z$ have elements $K_{x_{(i,j)}} = k_x(\mathbf{x}_i, \mathbf{x}_j)$ and $K_{z_{(i,j)}} = k_z(\mathbf{z}_i, \mathbf{z}_j)$, while $H = \mathbf{I} - \frac{1}{n}\mathbf{1}\mathbf{1}^\top$ is the centering matrix.

**Notation Summary.** Table 6 summarizes the notation used in the main paper and Appendix.

# B    H-SPLID Pseudocode

Algorithm 1 contains pseudo-code for H-SPLID, i.e., the alternating optimization algorithm presented in Section 3.4 to solve Problem (6).

Table 6: Summary of Notation and Terminology

| Symbol | Description |
|---|---|
| **Domains and Variables** | |
| $\mathcal{X} \subseteq \mathbb{R}^d$ | Input domain (bounded subset of $\mathbb{R}^d$) |
| $\mathcal{R} \subseteq \mathbb{R}^m$ | Representation domain (output of encoder) |
| $\mathcal{Z} \subseteq \mathbb{R}^m$ | Salient representation domain (output of encoder) |
| $\mathbb{R}^k$ | Output/logit space |
| $X \sim \mathcal{N}_R(0, \sigma^2 I_d)$ | Truncated multivariate normal on ball of radius $R$ |
| $Z := h_{\boldsymbol{\theta}}(\mathbf{x})$ | Output representation of the network |
| **Functions and Network Components** | |
| $\pi_i(\mathbf{x}) := x_i$ | $i$-th coordinate projection function |
| $f_{\boldsymbol{\psi}} : \mathcal{X} \to \mathcal{R}$ | $L$-Lipschitz encoder network |
| $\mathbf{M} \in \mathbb{R}^{m \times m}$ | Diagonal binary mask matrix (entries in $\{0, 1\}$) |
| $s := \mathrm{tr}(\mathbf{M})$ | Number of active (nonzero) dimensions in the mask |
| $\mathbf{W} \in \mathbb{R}^{k \times m}$ | Linear weight matrix after masking |
| $g_{\mathbf{W}} : \mathcal{Z} \to \mathbb{R}^k$ | Final representation-domain function (e.g., linear layer) |
| $h_{\boldsymbol{\theta}} := g_{\mathbf{W}} \circ f_{\boldsymbol{\psi}}$ | Full neural network from input to output |
| **Norms and Constants** | |
| $\|\cdot\|_2$ | Euclidean (vector) norm |
| $\|\cdot\|_F$ | Frobenius norm for matrices |
| $\|\cdot\|_\infty$ | Maximum absolute value across components (for vectors) |
| $\|\cdot\|_{\infty,2}$ | Supremum of 2-norm: $\|f\|_{\infty,2} := \sup_{\mathbf{x} \in \mathcal{X}} \|f(\mathbf{x})\|_2$ |
| $B := \|\mathbf{W}\|_\infty$ | Max row-wise $\ell_1$ norm of the weight matrix $\mathbf{W}$ |
| $N_{\mathcal{X}} := R$ | Sup-2-norm bound of input: $\|x\|_2 \le R$ |
| $N_{\mathcal{Z}} := B\sqrt{ks}(LR + \|f_{\boldsymbol{\psi}}(0)\|_2)$ | Sup-2-norm bound on $g_{\mathbf{W}} \circ f_{\boldsymbol{\psi}}$ |
| $\|\delta\|_2 \le r$ | Perturbation is bounded by $r$ |
| **Kernel and RKHS Quantities** | |
| $k_{\mathcal{X}}, k_{\mathcal{Z}}$ | Universal kernels on input and representation domains |
| $\mathcal{F}, \mathcal{G}$ | RKHSs induced by $k_{\mathcal{X}}$ and $k_{\mathcal{Z}}$ |
| $K_{\mathcal{F}}, K_{\mathcal{G}}$ | Kernel Sup-2-norm bounds |
| $\mathrm{HSIC}(\mathbf{x}, \mathbf{z})$ | Hilbert-Schmidt Independence Criterion between $X$ and $\mathbf{z}$ |
| **Loss and Perturbation** | |
| $\delta \in \mathbb{R}^d$ | Input perturbation with $\|\delta\|_2 \le r$ |
| $\mathcal{L}(h_{\boldsymbol{\theta}}(\mathbf{x}), y)$ | Loss function for prediction and ground-truth $y$ |
| $L_{\mathcal{L}}$ | Lipschitz constant of the loss in its first argument |

## C  Proof of Theorem 3.2

We next show that the output of a masked neural network is uniformly bounded in sup-norm under standard Lipschitz and compactness conditions. This provides the foundation for connecting the model class to the kernel-bounded spaces introduced above.

**Lemma C.1** (Bounded NN with Saliency Space). *Let $\mathcal{X} \subseteq \{\mathbf{x} \in \mathbb{R}^d : \|\mathbf{x}\|_2 \le R\}$ be a bounded input space, and let $f_{\boldsymbol{\psi}} : \mathcal{X} \to \mathbb{R}^m$ be an $L$-Lipschitz encoder. Consider a network $h_{\boldsymbol{\theta}}(\mathbf{x}) := \mathbf{W}\mathbf{M}f_{\boldsymbol{\psi}}(\mathbf{x})$ where $\mathbf{W} \in \mathbb{R}^{k \times m}$ is a linear weight matrix satisfying $\|\mathbf{W}\|_\infty \le B$, and $\mathbf{M} \in \mathbb{R}^{m \times m}$ is a diagonal binary mask with at most $s$ nonzero entries. Then, the network output is bounded in sup-norm:*

$$\|h_{\boldsymbol{\theta}}\|_{\infty,2} := \sup_{\mathbf{x} \in \mathcal{X}} \|h_{\boldsymbol{\theta}}(\mathbf{x})\|_2 \le B\sqrt{ks}(LR + \|f_{\boldsymbol{\psi}}(\mathbf{0})\|_2). \tag{18}$$

The proof is deferred to Appendix D.1. This bound shows that the sparsity level $s$ of the mask plays a direct role in constraining the model's output magnitude, which is essential for robustness.

To connect neural network outputs to kernel-based function spaces, we reparameterize the neural network $h_{\boldsymbol{\theta}}(\cdot) := \mathbf{W}\mathbf{M}_s f_{\boldsymbol{\psi}}(\cdot)$ by $g_{\mathbf{W}}(\mathbf{z}_s) \equiv \mathbf{W}\mathbf{z}_s$. Then, we show how the $g_{\mathbf{W}}$ belongs to a bounded subset of $\mathcal{C}(\mathcal{Z}, \mathbb{R}^k)$.

**Corollary C.2** (Bounded Function Spaces). *By redefining the neural network in Lemma C.1, $g_{\mathbf{W}}$ belongs to the following closed ball:*

$$g_{\mathbf{W}} \in \mathcal{C}_b^{N_{\mathcal{Z}}} := \left\{ g \in \mathcal{C}(\mathcal{Z}, \mathbb{R}^k) \,\middle|\, \|g\|_{\infty,2} \le N_{\mathcal{Z}} \equiv B\sqrt{ks}(LR + \|f_{\boldsymbol{\psi}}(\mathbf{0})\|_2) \right\}, \qquad (19)$$

*where $\|g\|_{\infty,2} := \sup_{\mathbf{z} \in \mathcal{Z}} \|g(\mathbf{z})\|_2$ denotes the sup-2-norm.*

This corollary imposes uniform boundedness of the neural network output values via $g_{\mathbf{W}}$ on representations over the compact domain $\mathcal{X}$, ensuring that the function belongs to a bounded subset of continuous function spaces $\mathcal{C}_b^{N_{\mathcal{Z}}}$. See Appendix D.2 for the proof.

Given the RKHSs $\mathcal{F}$ and $\mathcal{G}$ in Assumption 3.1, we define the rescaled RKHS spaces $\hat{\mathcal{F}}$ and $\hat{\mathcal{G}}$ as

$$\hat{\mathcal{F}} := \left\{ \frac{N_{\mathcal{X}}}{K_{\mathcal{F}}} f : f \in \mathcal{F} \right\} \quad and \quad \hat{\mathcal{G}} := \left\{ \frac{N_{\mathcal{Z}}}{K_{\mathcal{G}}} f : f \in \mathcal{G} \right\}. \qquad (20)$$

Thus, we establish the equivalence between the rescaled RKHSs and the bounded continuous function spaces.

**Lemma C.3** (Rescaled RKHS Equals $\mathcal{C}_b(\mathcal{X}, \mathbb{R}^d)$). *Given Assumption 3.1 and the continuous universal kernel $k_x$ therein, its corresponding RKHS $\mathcal{F}$, and a bounded continuous function space $\mathcal{C}_b(\mathcal{X}, \mathbb{R}^d)$ such that*

$$\mathcal{C}_b^{N_{\mathcal{X}}} := \left\{ f \in \mathcal{C}(\mathcal{X}, \mathbb{R}^d) \,\middle|\, \|f\|_{\infty,2} \le N_{\mathcal{X}} \right\},$$

*then we have the rescaled RKHS space*

$$\hat{\mathcal{F}} := \left\{ \frac{N_{\mathcal{X}}}{K_{\mathcal{F}}} f : f \in \mathcal{F} \right\} \qquad (21)$$

*satisfying*

$$\overline{\mathcal{F}} = \mathcal{C}_b^{N_{\mathcal{X}}}. \qquad (22)$$

*where $\overline{\mathcal{F}} := \overline{\hat{\mathcal{F}}}^{\|\cdot\|_{\infty,2}}$ denotes the closure of $\hat{\mathcal{F}}$ w.r.t. the $\|\cdot\|_{\infty,2}$.*

Similarly, we can show that the rescaled $\mathcal{C}_b^{N_{\mathcal{Z}}} = \overline{\hat{\mathcal{G}}}^{\|\cdot\|_{\infty,2}}$. See Appendix D.3 for the proof.

Moreover, as we have two spaces containing functions that are different by a scalar, we are interested in how the supremums of the covariance relate between spaces.

**Lemma C.4** (Scaling of Supremum Covariance — Sum Version). *Let $\mathcal{F}$ and $\mathcal{G}$ be vector-valued RKHSs over $\mathcal{X}$ and $\mathcal{Z}$, respectively. Then, for all $M_{\mathcal{F}}, M_{\mathcal{G}} > 0$ and $\mathbf{x} \in \mathcal{X}, \mathbf{z}_s \in \mathcal{Z}$, the following holds:*

$$\sum_{j=1}^{k} \sum_{i=1}^{d} \sup_{f_j \in \mathcal{F}_j, g_i \in \mathcal{G}_i} \mathrm{Cov}[f_j(\mathbf{x}), g_i(\mathbf{z}_s)] = M_{\mathcal{F}} M_{\mathcal{G}} \sum_{j=1}^{k} \sum_{i=1}^{d} \sup_{\tilde{f}_j \in \hat{\mathcal{F}}_j, \tilde{g}_i \in \hat{\mathcal{G}}_i} \mathrm{Cov}[\tilde{f}_j(\mathbf{x}), \tilde{g}_i(\mathbf{z}_s)], \quad (23)$$

*where $\hat{\mathcal{F}} := \{\tilde{f} = f/M_{\mathcal{F}} : f \in \mathcal{F}\}$ and likewise for $\hat{\mathcal{G}}$.*

See Appendix D.4 for the proof.

Lastly, given the supremum of covariance of the function space containing the neural network, we need to use the variant of Stein's Lemma to bound the gradient of the neural network.

**Lemma C.5** (Stein's Lemma for Scalar-Valued Functions on a Bounded Domain). *Let $\mathbf{x}$ be sampled from a truncated multivariate normal (tMVN) distribution, i.e., with density $\tilde{p}(\mathbf{x}) = \frac{1}{C} \exp\left(-\frac{\|\mathbf{x}\|^2}{2\sigma^2}\right) \mathbf{1}_{\|x\| \le R}$, supported on the compact domain $\mathcal{X}_R := \{\mathbf{x} \in \mathbb{R}^d : \|\mathbf{x}\| \le R\}$, where $C$ is the normalization constant. Let $h : \mathbb{R}^d \to \mathbb{R}$ be a differentiable almost everywhere such that $\mathbb{E}|\partial h(\mathbf{x})/\partial x_i| < \infty$ and $|h(\mathbf{x})| \le N_{\mathcal{X}}$ for all $x \in \mathcal{X}_R$. Then for all $i \in \{1, \dots, d\}$,*

$$\mathbb{E}\left[\frac{\partial h(\mathbf{x})}{\partial x_i}\right] = \frac{1}{\sigma^2} \mathrm{Cov}\left[x_i, h(\mathbf{x})\right] + \epsilon_i(R), \qquad (24)$$

*where the truncation error term satisfies*

$$|\epsilon_i(R)| \leq \frac{N_{\mathcal{X}} C_d R^{d-1}}{C} \exp\left(-\frac{R^2}{2\sigma^2}\right),$$

*and $C_d$ is the surface area of the unit sphere in $\mathbb{R}^d$.*

See Appendix D.5 for the proof.

Then, we formally state the proof of Theorem 3.2.

*Proof.* **Step 1. Continuous function spaces $C_{\mathcal{X}}^{N_{\mathcal{X}}}$ and $C_{\mathcal{Z}}^{N_{\mathcal{Z}}}$.**

Let $\pi : \mathcal{X} \to \mathbb{R}^d$ denote the identity map, defined by $\pi(\mathbf{x}) := \mathbf{x}$. This vector-valued function can be decomposed into scalar coordinate projections:

$$\pi_i(\mathbf{x}) := x_i, \quad \text{for } i = 1, \ldots, d.$$

Since the input domain $\mathcal{X} \subseteq \mathbb{R}^d$ is contained within a Euclidean ball of radius $R$, we have $\|\pi(\mathbf{x})\|_2 \leq R$ for all $x \in \mathcal{X}$. Therefore, the identity function satisfies:

$$N_{\mathcal{X}} := \|\pi\|_{\infty,2} = R,$$

and lies in the vector-valued bounded continuous function space $\pi \in \mathcal{C}_b^{N_{\mathcal{X}}}(\mathcal{X}, \mathbb{R}^d)$. Correspondingly, each coordinate function belongs to the subspace $\pi_i \in C_{b,i}^{N_{\mathcal{X}}}$.

Now consider the function $g_{\mathbf{W}} : \mathcal{Z} \to \mathbb{R}^k$ on the representation domain $\mathcal{Z}$. From Corollary C.2, the composed function $g_{\mathbf{W}} \circ \mathbf{M}_s f_{\boldsymbol{\psi}}$ over $\mathcal{X}$ satisfies:

$$N_{\mathcal{Z}} := B\sqrt{ks}(LR + \|f_{\boldsymbol{\psi}}(0)\|_2).$$

Similarly, over the representation space $\mathcal{Z}$, we have $g_{\mathbf{W}} \in \mathcal{C}_b^{N_{\mathcal{Z}}}(\mathcal{Z}, \mathbb{R}^k)$, and each scalar component $g_{\mathbf{W}}^{(j)} \in C_{b,j}^{N_{\mathcal{Z}}}$.

**Step 2. Equivalence between RKHS and continous function spaces.**

By Lemma C.3, we can rescale the RKHS $\mathcal{F}$ and $\mathcal{G}$ in Assumption 3.1 as

$$\hat{\mathcal{F}} := \left\{ \frac{N_{\mathcal{X}}}{K_{\mathcal{F}}} f : f \in \mathcal{F} \right\} \quad \text{and} \quad \hat{\mathcal{G}} := \left\{ \frac{N_{\mathcal{Z}}}{K_{\mathcal{G}}} g : g \in \mathcal{G} \right\}, \tag{25}$$

so that their closure are equivalent to the bounded continuous function space $C_{\mathcal{X}}^{N_{\mathcal{X}}}$ and $C_{\mathcal{Z}}^{N_{\mathcal{Z}}}$ as in step 1.

According to Lemma C.4, if we set $M_{\mathcal{F}} := \frac{K_{\mathcal{F}}}{N_{\mathcal{X}}}, M_{\mathcal{G}} := \frac{K_{\mathcal{G}}}{N_{\mathcal{Z}}}$ we relate covariance bounds between the RKHSs ($\mathcal{F}$ and $\mathcal{G}$) and the rescaled RKHSs ($\hat{\mathcal{F}}$ and $\hat{\mathcal{G}}$) through rescaling.

$$\sum_{j=1}^{k} \sum_{i=1}^{d} \sup_{f_j \in \mathcal{F}_j, g_i \in \mathcal{G}_i} \text{Cov}[f_j(\mathbf{x}), g_i(\mathbf{z}_s)] \tag{26}$$

$$= \frac{K_{\mathcal{F}} K_{\mathcal{G}}}{N_{\mathcal{X}} N_{\mathcal{Z}}} \sum_{j=1}^{k} \sum_{i=1}^{d} \sup_{\hat{f}_j \in \hat{\mathcal{F}}_j, \hat{g}_i \in \hat{\mathcal{G}}_i} \text{Cov}[\hat{f}_j(\mathbf{x}), \hat{g}_i(\mathbf{z}_s)] \qquad \text{(Lemma C.4)}$$

$$= \frac{K_{\mathcal{F}} K_{\mathcal{G}}}{N_{\mathcal{X}} N_{\mathcal{Z}}} \sum_{j=1}^{k} \sum_{i=1}^{d} \sup_{\hat{f}_j \in \overline{\hat{\mathcal{F}}}_j, \hat{g}_i \in \overline{\hat{\mathcal{G}}}_i} \text{Cov}[\hat{f}_j(\mathbf{x}), \hat{g}_i(\mathbf{z}_s)] \qquad \text{(closure under sup-norm)}$$

$$= \frac{K_{\mathcal{F}} K_{\mathcal{G}}}{N_{\mathcal{X}} N_{\mathcal{Z}}} \sum_{j=1}^{k} \sum_{i=1}^{d} \sup_{\tilde{f}_j \in C_{b,j}^{N_{\mathcal{X}}}, \tilde{g}_i \in C_{b,i}^{N_{\mathcal{Z}}}} \text{Cov}[\tilde{f}_j(\mathbf{x}), \tilde{g}_i(\mathbf{z}_s)], \qquad \text{(Lemma C.3)}$$

where the second last line applies closure under the sup-norm (preserving the supremum), and the last line substitutes the equivalent bounded continuous function space by Lemma C.3.

## Step 3. Bound covariance in continuous function spaces with HSIC.

Then, based on Eq. (16), we obtain:

$$\sum_{j=1}^{k}\sum_{i=1}^{d} \sup_{f_j \in \mathcal{F}_j, g_i \in \mathcal{G}_i} \text{Cov}[f_j(\mathbf{x}), g_i(\mathbf{z}_s)] \leq \text{HSIC}(\mathbf{x}, \mathbf{z}_s). \tag{27}$$

Combining eq. (26) and eq. (27), we have

$$\sum_{j=1}^{k}\sum_{i=1}^{d} \sup_{\tilde{f}_j \in C_{b,j}^{N_{\mathcal{X}}}, \hat{g}_i \in C_{b,i}^{N_{\mathcal{Z}}}} \text{Cov}[\tilde{f}_j(\mathbf{x}), \tilde{g}_i(\mathbf{z}_s)] \leq \frac{N_{\mathcal{X}} N_{\mathcal{Z}}}{K_{\mathcal{F}} K_{\mathcal{G}}} \cdot \text{HSIC}(\mathbf{x}, \mathbf{z}_s). \tag{28}$$

As shown in Step 1, we have $\pi_i \in C_{b,i}^{N_{\mathcal{X}}}, h_{\boldsymbol{\theta}}^{(j)} \in C_{b,j}^{N_{\mathcal{Z}}}$, and the following holds

$$\sum_{j=1}^{k}\sum_{i=1}^{d} \sup_{\pi_i \in C_{b,i}^{N_{\mathcal{X}}}, h_{\boldsymbol{\theta}}^{(j)} \in C_{b,j}^{N_{\mathcal{Z}}}} \text{Cov}[\pi_i(\mathbf{x}), h_{\boldsymbol{\theta}}^{(j)}(\mathbf{x})] \leq \frac{N_{\mathcal{X}} N_{\mathcal{Z}}}{K_{\mathcal{F}} K_{\mathcal{G}}} \cdot \text{HSIC}(\mathbf{x}, \mathbf{z}_s). \tag{29}$$

## Step 4. Bound the gradient with covariance.

By Lemma C.5, the following holds

$$\left| \mathbb{E}\left[ \frac{\partial h_{\boldsymbol{\theta}}^{(j)}(\mathbf{x})}{\partial x_i} \right] \right| = \frac{1}{\sigma^2} \left| \text{Cov}[X_i, h_{\boldsymbol{\theta}}^{(j)}(\mathbf{x})] \right| \leq \frac{1}{\sigma^2} \sup_{\pi_i \in C_{b,i}^{N_{\mathcal{X}}}, h_{\boldsymbol{\theta}}^{(j)} \in C_{b,j}^{N_{\mathcal{Z}}}} \text{Cov}[\pi_i(\mathbf{x}), h_{\boldsymbol{\theta}}^{(j)}(\mathbf{x})]. \tag{30}$$

Combining (29) and (30) gives:

$$\sum_{j=1}^{k}\sum_{i=1}^{d} \left| \mathbb{E}\left[ \frac{\partial h_{\boldsymbol{\theta}}^{(j)}(\mathbf{x})}{\partial x_i} \right] \right| \leq \frac{1}{\sigma^2} \left( \frac{N_{\mathcal{X}} N_{\mathcal{Z}}}{K_{\mathcal{F}} K_{\mathcal{G}}} \cdot \text{HSIC}(\mathbf{x}, \mathbf{z}_s) + \epsilon_R \right). \tag{31}$$

Consider the first-order Taylor expansion in a Euclidian ball of radius $r$ around $\mathbf{x} \in \mathcal{X}$: that is,

$$h_{\boldsymbol{\theta}}(\mathbf{x} + \delta) - h_{\boldsymbol{\theta}}(\mathbf{x}) = J_{h_{\boldsymbol{\theta}}}(\mathbf{x})\delta + o(r). \tag{32}$$

for all $\mathbf{x} \in \mathcal{X}$ and all $\delta \in \mathbb{R}^d$ s.t. $\|\delta\|_2 \leq r$.

Consider now a measurable perturbation function on $\mathcal{X}$ as $\delta : \mathbb{R}^d \to \mathbb{R}^d$, such that

$$\sup_{\mathbf{x} \in \mathcal{X}} \|\delta(\mathbf{x})\|_2 \leq r. \tag{33}$$

As Eq. (32) holds for all $\mathbf{x}, \delta$ pairs, for all $\mathbf{x} \in \mathcal{X}$, we have that:

$$h_{\boldsymbol{\theta}}(\mathbf{x} + \delta(\mathbf{x})) - h_{\boldsymbol{\theta}}(\mathbf{x}) = J_{h_{\boldsymbol{\theta}}}(\mathbf{x})\delta(\mathbf{x}) + o(r). \tag{34}$$

We thus have that, for all $\mathbf{x} \in \mathcal{X}$:

$$\left\| h_{\boldsymbol{\theta}}(\mathbf{x} + \delta(\mathbf{x})) - h_{\boldsymbol{\theta}}(\mathbf{x}) \right\|_2 \leq \left\| J_{h_{\boldsymbol{\theta}}}(\mathbf{x})\, \delta(\mathbf{x}) \right\|_2 + o(r) \tag{35}$$

$$\leq \left\| J_{h_{\boldsymbol{\theta}}}(\mathbf{x}) \right\|_F \left\| \delta(\mathbf{x}) \right\|_2 + o(r) \qquad \text{by Cauchy-Schwartz,} \tag{36}$$

$$\leq r \cdot \left\| J_{h_{\boldsymbol{\theta}}}(\mathbf{x}) \right\|_F + o(r) \qquad \text{by Eq. (33).} \tag{37}$$

Hence, it follows that:

$$\mathbb{E}\left[ \left\| h_{\boldsymbol{\theta}}(X + \delta(X)) - h_{\boldsymbol{\theta}}(X) \right\|_2 \right] \leq r \cdot \mathbb{E}\left[ \left\| J_{h_{\boldsymbol{\theta}}}(X) \right\|_F \right] + o(r) \tag{38}$$

$$\leq \frac{r R B \sqrt{k}s(LR + \|f_{\boldsymbol{\psi}}(0)\|_2)}{\sigma^2 K_{\mathcal{F}} K_{\mathcal{G}}} \cdot \text{HSIC}(X, \mathbf{z}_s) + o(r), \tag{39}$$

where the last inequality follows from Eq. (31) and the fact $\frac{\epsilon_R}{\sigma^2} = o(1)$, due to the exponentially decaying term. $\qquad\square$

# D  Proof of Lemmas used in Theorem 3.2

## D.1  Proof of Lemma C.1

*Proof.* Let $\mathbf{W}_n \in \mathbb{R}^{k \times s}$ and $\mathbf{M}_s \in \mathbb{R}^{s \times m}$ be the pruned matrices selecting the active coordinates corresponding to the $s$ nonzero entries of the mask. Then the function can equivalently be rewritten as

$$h(\mathbf{x}) = \mathbf{W}_s \mathbf{M}_s f_\psi(\mathbf{x}). \tag{40}$$

Since $f_\psi$ is $L$-Lipschitz and $\|\mathbf{x}\|_2 \leq R$, it follows that

$$\|f_\psi(\mathbf{x}) - f_\psi(\mathbf{0})\|_2 \leq LR, \tag{41}$$

thus

$$\|f_\psi(\mathbf{x})\|_2 \leq \|f_\psi(\mathbf{0})\|_2 + LR, \quad \forall \mathbf{x} \in \mathcal{X}. \tag{42}$$

The masking operation $\mathbf{M}$ selects $s$ coordinates from $f_\psi(\mathbf{x})$, and can be equivalently represented via $\mathbf{M}_s \in \mathbb{R}^{s \times m}$ as a selector matrix with exactly one nonzero entry per row and no more than one nonzero per column. Then

$$\|\mathbf{M}_s f_\psi(\mathbf{x})\|_2 \leq \|f_\psi(\mathbf{x})\|_2. \tag{43}$$

The corresponding reduced weight matrix $\mathbf{W}_s \in \mathbb{R}^{k \times s}$ selects the columns of $\mathbf{W}$ associated with the active coordinates. Since $\|\mathbf{W}_{ij}\|_\infty \leq B$, it follows that

$$\|\mathbf{W}_s\|_F \leq B\sqrt{ks}, \tag{44}$$

and thus

$$\|\mathbf{W}_s\|_{2\to2} \leq \|\mathbf{W}_s\|_F \leq B\sqrt{ks}. \tag{45}$$

where $\|\cdot\|_F$ is the Frobenius norm. Hence, for any $\mathbf{x} \in \mathcal{X}$,

$$\|h(\mathbf{x})\|_2 = \|\mathbf{W}_s \mathbf{M}_s f_\psi(\mathbf{x})\|_2 \leq \|\mathbf{W}_s\|_{2\to2} \|\mathbf{M}_s f_\psi(\mathbf{x})\|_2 \leq B\sqrt{ks}(LR + \|f_\psi(\mathbf{0})\|_2). \tag{46}$$

Taking the supremum over $\mathbf{x} \in \mathcal{X}$ concludes the proof. □

## D.2  Proof of Corollary C.2

*Proof.* As we have shown in Lemma C.1, we have:

$$\sup_{\mathbf{x} \in \mathcal{X}} \|h(\mathbf{x})\|_2 \leq B\sqrt{ks}(LR + \|f_\psi(\mathbf{0})\|_2).$$

Moreover, since the NN can be expressed as $h_\theta(\cdot) = \mathbf{W}\mathbf{M}f_\psi(\cdot)$, we have:

$$\sup_{\mathbf{x} \in \mathcal{X}} \|\mathbf{W}\mathbf{M}f_\psi(\mathbf{x})\|_2 \leq B\sqrt{ks}(LR + \|f_\psi(\mathbf{0})\|_2).$$

Therefore, we can upper bound $g_{\mathbf{w}}$ as:

$$\sup_{\mathbf{z} \in \mathcal{Z}} \|g_{\mathbf{w}}(\mathbf{z})\|_2 = \sup_{\mathbf{z} \in \mathcal{Z}} \|\mathbf{W}\mathbf{M}\mathbf{z}\|_2 \leq B\sqrt{ks}(LR + \|f_\psi(\mathbf{0})\|_2).$$

Moreover, as $g_{\mathbf{W}}$ is a continuous function, we finish the proof. □

## D.3  Proof of Lemma C.3

*Proof.* **Step 1.** $\overline{\mathcal{F}} \subseteq \mathcal{C}_b^{N_\mathcal{X}}$

Since $k_x$ is continuous and $\mathcal{X}$ is compact, it follows from Lemma 4.28 of Steinwart and Christmann [47] that all $f \in \mathcal{F}$ are bounded and continuous. Hence, for any $f \in \mathcal{F}$, we have

$$\left\|\frac{N_\mathcal{X}}{K_\mathcal{F}}f\right\|_{\infty,2} = \frac{N_\mathcal{X}}{K_\mathcal{F}}\|f\|_{\infty,2} \leq N_\mathcal{X}. \tag{47}$$

This implies that every function in $\hat{\mathcal{F}}$ belongs to $\mathcal{C}_b^{N_{\mathcal{X}}}$. Since $\mathcal{C}_b^{N_{\mathcal{X}}}$ is closed in the $\|\cdot\|_{\infty,2}$ norm, it follows that

$$\overline{\mathcal{F}} \subseteq \mathcal{C}_b^{N_{\mathcal{X}}}. \tag{48}$$

**Step 2.** $\mathcal{C}_b^{N_{\mathcal{X}}} \subseteq \overline{\mathcal{F}}$

Let $g \in \mathcal{C}_b^{N_{\mathcal{X}}}$. Define $h := \frac{K_{\mathcal{F}}}{N_{\mathcal{X}}} g$. Then,

$$\|h\|_{\infty,2} = \frac{K_{\mathcal{F}}}{N_{\mathcal{X}}} \|g\|_{\infty,2} \leq K_{\mathcal{F}}, \tag{49}$$

so $h \in \mathcal{C}_b(\mathcal{X}, \mathbb{R}^k)$ and is bounded in sup-norm. Since $\mathcal{F}$ is universal by Assumption. 3.1, it is dense in $\mathcal{C}_b(\mathcal{X}, \mathbb{R}^k)$ under the $\|\cdot\|_{\infty,2}$ norm. Therefore, there exists a sequence $\{f_n\} \subset \mathcal{F}$ such that

$$\lim_{n \to \infty} \|f_n - h\|_{\infty,2} = 0. \tag{50}$$

Define the corresponding rescaled sequence $\hat{f}_n := \frac{N_{\mathcal{X}}}{K_{\mathcal{F}}} f_n \in \hat{\mathcal{F}}$, and set $\hat{g} := \frac{N_{\mathcal{X}}}{K_{\mathcal{F}}} h = g$. Computing the limit,

$$\lim_{n \to \infty} \|\hat{f}_n - \hat{g}\|_{\infty,2} = \lim_{n \to \infty} \left\| \frac{N_{\mathcal{X}}}{K_{\mathcal{F}}} (f_n - h) \right\|_{\infty,2} = \lim_{n \to \infty} \frac{N_{\mathcal{X}}}{K_{\mathcal{F}}} \|f_n - h\|_{\infty,2} = 0. \tag{51}$$

Thus, $g = \hat{g} \in \overline{\mathcal{F}}$. $\qquad\square$

### D.4 Proof of Lemma C.4

*Proof.* We show that

$$\sum_{j,i} \sup_{f_j,g_i} \mathrm{Cov}[f_j(\mathbf{x}), g_i(\mathbf{z}_s)] \leq M_{\mathcal{F}} M_{\mathcal{G}} \sum_{j,i} \sup_{\tilde{f}_j,\tilde{g}_i} \mathrm{Cov}[\tilde{f}_j(\mathbf{x}), \tilde{g}_i(\mathbf{z}_s)],$$

For each pair $(j, i)$, let $\{f_n^{(j)}\} \subset \mathcal{F}_j$ and $\{g_n^{(i)}\} \subset \mathcal{G}_i$ be sequences converging to the limit

$$\lim_{n \to \infty} \mathrm{Cov}[f_n^{(j)}(\mathbf{x}), g_n^{(i)}(\mathbf{z}_s)] = \sup_{f_j,g_i} \mathrm{Cov}[f_j(\mathbf{x}), g_i(\mathbf{z}_s)]. \tag{52}$$

Define the rescaled sequences:

$$\tilde{f}_n^{(j)} := \frac{1}{M_{\mathcal{F}}} f_n^{(j)}, \quad \tilde{g}_n^{(i)} := \frac{1}{M_{\mathcal{G}}} g_n^{(i)}. \tag{53}$$

Then, by the bilinearity of the covariance operator, we have

$$\mathrm{Cov}[f_n^{(j)}(\mathbf{x}), g_n^{(i)}(\mathbf{z}_s)] = M_{\mathcal{F}} M_{\mathcal{G}} \, \mathrm{Cov}[\tilde{f}_n^{(j)}(\mathbf{x}), \tilde{g}_n^{(i)}(\mathbf{z}_s)], \tag{54}$$

and taking the limit:

$$\sup_{f_j,g_i} \mathrm{Cov}[f_j(\mathbf{x}), g_i(\mathbf{z}_s)] = \lim_{n \to \infty} \mathrm{Cov}[f_n^{(j)}(\mathbf{x}), g_n^{(i)}(\mathbf{z}_s)] \tag{55}$$

$$= M_{\mathcal{F}} M_{\mathcal{G}} \lim_{n \to \infty} \mathrm{Cov}[\tilde{f}_n^{(j)}(\mathbf{x}), \tilde{g}_n^{(i)}(\mathbf{z}_s)] \leq M_{\mathcal{F}} M_{\mathcal{G}} \sup_{\tilde{f}_j,\tilde{g}_i} \mathrm{Cov}[\tilde{f}_j(\mathbf{x}), \tilde{g}_i(\mathbf{z}_s)]. \tag{56}$$

Summing over all $i, j$ yields the result.

Furthermore, the reverse inequality follows from the same argument. $\qquad\square$

### D.5 Proof of Lemma C.5

*Proof.* **Step 1 (Integration by Parts).** Let $\phi(\mathbf{x}) = \exp\left(-\frac{\|\mathbf{x}\|^2}{2\sigma^2}\right)$, and define $f(\mathbf{x}) = h(\mathbf{x})\phi(\mathbf{x})$. Applying the product rule:

$$\frac{\partial}{\partial x_i} f(\mathbf{x}) = \frac{\partial h(\mathbf{x})}{\partial x_i}\phi(\mathbf{x}) + h(\mathbf{x})\frac{\partial \phi(\mathbf{x})}{\partial x_i}. \tag{57}$$

Rearranging:

$$\frac{\partial h(\mathbf{x})}{\partial x_i}\phi(\mathbf{x}) = \frac{\partial}{\partial x_i}(h(\mathbf{x})\phi(\mathbf{x})) - h(\mathbf{x})\frac{\partial \phi(\mathbf{x})}{\partial x_i}. \tag{58}$$

Integrating over $\mathcal{X}_R$ and applying the divergence theorem gives:

$$\int_{\mathcal{X}_R} \frac{\partial}{\partial x_i}(h(\mathbf{x})\phi(\mathbf{x}))\, dx = \int_{\partial \mathcal{X}_R} h(\mathbf{x})\phi(\mathbf{x})\nu_i(\mathbf{x})\, dS(\mathbf{x}), \tag{59}$$

where $\nu(\mathbf{x}) = \frac{\mathbf{x}}{\|\mathbf{x}\|}$ is the outward unit normal and $\nu_i(\mathbf{x}) = \frac{x_i}{R}$.

Thus:

$$\int_{\mathcal{X}_R} \frac{\partial h(\mathbf{x})}{\partial x_i}\phi(\mathbf{x})\, dx = \int_{\partial \mathcal{X}_R} h(\mathbf{x})\phi(\mathbf{x})\nu_i(\mathbf{x})\, dS(\mathbf{x}) + \int_{\mathcal{X}_R} h(\mathbf{x})\frac{x_i}{\sigma^2}\phi(\mathbf{x})\, dx. \tag{60}$$

**Step 2 (Pass to Expectation Form).** Dividing through by $C = \int_{\|x\| \leq R} \phi(\mathbf{x})\, dx$, the normalization constant, gives:

$$\mathbb{E}\left[\frac{\partial h(\mathbf{x})}{\partial x_i}\right] = \frac{1}{\sigma^2}\mathbb{E}\left[X_i h(\mathbf{x})\right] + \frac{1}{C}\int_{\|x\|=R} h(\mathbf{x})\phi(\mathbf{x})\nu_i(\mathbf{x})\, dS(\mathbf{x}). \tag{61}$$

**Step 3 (Bounding the Boundary Term).** Since $|\nu_i(\mathbf{x})| \leq 1$ and $|h(\mathbf{x})| \leq \frac{N_{\mathcal{X}}}{\sqrt{k}}$ on $\mathcal{X}_R$, we have:

$$\left| \int_{\|x\|=R} h(\mathbf{x})\phi(\mathbf{x})\nu_i(\mathbf{x})\, dS(\mathbf{x}) \right| \leq \frac{N_{\mathcal{X}}}{\sqrt{k}} \exp\left(-\frac{R^2}{2\sigma^2}\right) \int_{\|x\|=R} dS(\mathbf{x}). \tag{62}$$

The surface area of the sphere is:

$$\int_{\|x\|=R} dS(\mathbf{x}) = C_d R^{d-1}, \tag{63}$$

where the constant $C_d > 0$ is the surface area of a d-dimensional unit sphere, depending only on $d$.

Thus, we can bound the error term as

$$|\epsilon_{i,j}(R)| \leq \frac{N_{\mathcal{X}} C_d R^{d-1}}{\sqrt{k}C} \exp\left(-\frac{R^2}{2\sigma^2}\right). \tag{64}$$

Then, as $R \to \infty$, we have:

$$kd \cdot \frac{N_{\mathcal{X}} C_d R^{d-1}}{\sqrt{k}C} \exp\left(-\frac{R^2}{2\sigma^2}\right) = o(1), \tag{65}$$

since the exponential decay dominates polynomial growth.

$\square$

## E   Proof of Lemma 3.3

*Proof.* Under the same assumptions as in Theorem 3.2, recall the HSIC bound on gradients in eq. (31) as

$$\sum_{j=1}^{k}\sum_{i=1}^{d}\left|\mathbb{E}\left[\frac{\partial h_{\boldsymbol{\theta}}^{(j)}(\mathbf{x})}{\partial x_i}\right]\right| \leq \frac{1}{\sigma^2}\left(\frac{N_{\mathcal{X}}N_{\mathcal{Z}}}{K_{\mathcal{F}}K_{\mathcal{G}}} \cdot \text{HSIC}(\mathbf{x}, \mathbf{z}_s) + \epsilon_R\right). \tag{66}$$

Table 7: License and source compliance for each dataset.

| Dataset | URL | License |
|---|---|---|
| ImageNet-1K [11] | image-net.org | ImageNet Terms |
| ImageNet-9 [58] | GitHub | Inherits ImageNet Terms |
| COCO [28] | cocodataset.org | CC BY 4.0 (annotations) / Flickr TOU (images) |
| CounterAnimal [55] | counteranimal.github.io | Inherits iNaturalist Terms |
| ISIC-2017 [9] | ISIC Challenge | CC-0 |
| C-MNIST | (our codebase) | Inherits MNIST Terms (CC BY-SA 3.0) |

Thus, we can bound the Frobenius norm of the gradient as

$$\mathbb{E}\|\nabla_{\mathbf{x}} h_{\boldsymbol{\theta}}(\mathbf{x})\|_F \leq \sum_{j=1}^{k} \sum_{i=1}^{d} \left| \mathbb{E}\left[ \frac{\partial h_{\boldsymbol{\theta}}^{(j)}(\mathbf{x})}{\partial x_i} \right] \right| \leq \frac{1}{\sigma^2} \left( \frac{N_{\mathcal{X}} N_{\mathcal{Z}}}{K_{\mathcal{F}} K_{\mathcal{G}}} \cdot \text{HSIC}(\mathbf{x}, \mathbf{z}_s) + \epsilon_R \right). \quad (67)$$

By Markov inequality, we have

$$\mathbb{P}(\|\nabla_{\mathbf{x}} h_{\boldsymbol{\theta}}(\mathbf{x})\|_F > \epsilon) \leq \frac{1}{\epsilon} \mathbb{E}\left[ \|\nabla_{\mathbf{x}} h_{\boldsymbol{\theta}}(\mathbf{x})\|_F \right] \quad (68)$$

Thus, plugging eq. (67) to eq. (68), we have

$$\mathbb{P}(\|\nabla_{\mathbf{x}} h_{\boldsymbol{\theta}}(\mathbf{x})\|_F > \epsilon) \leq \frac{1}{\epsilon \sigma^2} \left( \frac{N_{\mathcal{X}} N_{\mathcal{Z}}}{K_{\mathcal{F}} K_{\mathcal{G}}} \cdot \text{HSIC}(\mathbf{x}, \mathbf{z}_s) + \epsilon_R \right). \quad (69)$$

As the error term $\epsilon_R = O(e^{-\frac{R^2}{2\sigma^2}})$, we have

$$\mathbb{P}(\|\nabla_{\mathbf{x}} h_{\boldsymbol{\theta}}(\mathbf{x})\|_F > \epsilon) \leq \frac{1}{\epsilon \sigma^2} \left( \frac{N_{\mathcal{X}} N_{\mathcal{Z}}}{K_{\mathcal{F}} K_{\mathcal{G}}} \cdot \text{HSIC}(\mathbf{x}, \mathbf{z}_s) \right) + o(1). \quad (70)$$

Substituting the $N_{\mathcal{X}}, N_{\mathcal{Z}}$, we finish the proof. $\qquad \square$

## F  Reproducibility Details

### F.1  Datasets

COCO is a segmentation dataset consisting of labeled images of various species of animals (See Figure 3b). For our experiments, we utilize a subset of the dataset composed of images drawn from one of four labels. The four species were carefully selected to ensure the largest possible dataset containing images without overlapping labels. Since we use the dataset for image classification, each sample should belong to one class and thus include animals from one and only one of the four selected classes. During pre-processing, the dataset is resized to 224x224 pixels. Finally, segmentation information is used to construct 224x224 masks, where the 0 entries denote the pixels occupied by the animal (salient object) in the original image. These masks specify the portion of the image shielded from adversarial perturbations. The splits are created from the public training data of COCO by splitting them into train (4509 samples), validation (1127 samples) and test (1411 samples).

C-MNIST is a synthetically constructed variant of the original MNIST dataset [27]. To generate it, we first load the standard 28x28 single-channel digit images. Subsequently, each sample is randomly paired with another digit using a fixed seed for reproducibility. The two images are concatenated along the width to form a 56x28 composite, then symmetrically zero-padded to a uniform 64x64 resolution. During training and evaluation, we treat only the left-hand digit as the classification target, ensuring each composite image belongs to exactly one class. We use 80% of the original train split of MNIST as training data and 20% as validation data. For testing we use the test set of MNIST, where we also create image pairs as described above.

To assess whether H-SPLID attends preferentially to salient objects rather than background cues, we evaluate it on two complementary benchmarks: (1) CounterAnimal (CA) [55], which splits iNaturalist

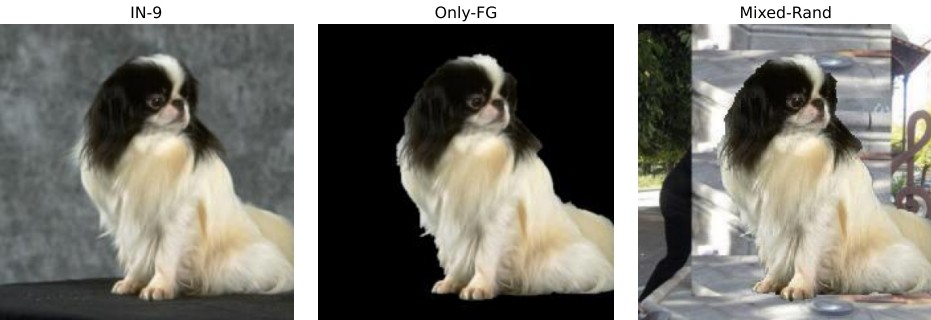

Figure 4: Samples from the three ImageNet9 variations: IN-9 (original), Only-FG, and Mixed-Rand.

wildlife photos into a Common set (exhibiting typical backgrounds) and a Counter set (featuring atypical yet plausible backgrounds, see Figure 3a), and (2) ImageNet-9 (IN-9) [58], defined as a subset of ImageNet-1K consisting of 368 categories, instantiated in three distinct variants: *Original* images, a *MixedRand* variant in which object foregrounds are transposed onto random-class backgrounds, and an *Only-FG* variant with backgrounds entirely removed (See Figure 4).

Table 7 provides each dataset's source URL and applicable licensing terms.

### F.2 Implementation

The HBaR code was adapted from the original codebase, which is publicly available at GitHub (under MIT License). For weight decay, we reuse the PyTorch [39] implementation and pass it directly to the optimizer. We re-implemented the following regularization methods: Group Lasso Weights, Group Lasso Activations, L1 Sparse Activations, and L1 Sparse Weights. For both the Projected Gradient Descent (PGD) [34] and AutoAttack (AA) [10] adversaries, we utilize our version of the TorchAttacks [23] library, that is adapted for masked attacks.

### F.3 Software and Hardware Setup

We built our pipeline in Python, leveraging the PyTorch [39] library. To conduct our experiments, we use two identical internal servers running Ubuntu 22.04.3 LTS ("Jammy Jellyfish") on a 5.15.0-84 x86_64 kernel. Each server is equipped with two Intel Xeon Gold 6326 processors (16 cores each, hyper-threaded for a total of 64 logical CPUs), 512 GiB of RAM, and a single NVIDIA A100 80 GB GPU. For the ablation studies and experiments conducted on the ISIC-2017 dataset, we additionally made use of EuroHPC compute resources, including MareNostrum (BSC, Spain), MeluXina (LuxProvide, Luxembourg), Deucalion (MACC, Portugal), and Discoverer (Sofia Tech, Bulgaria).

### F.4 Hyperparameters

We divide our hyperparameters into two groups: those shared by all models, and those tuned or adapted per method and dataset.

**Shared parameters.** All ImageNet-1K ResNet-50 and COCO ResNet-18 experiments use the Adam optimizer [24] with $\beta_1 = 0.9, \beta_2 = 0.999, \epsilon = 10^{-8}$. We perform an initial grid search on a vanilla ResNet-18, sweeping the learning rate over $\{10^{-3}, 10^{-4}, 10^{-5}, 10^{-6}\}$ in logarithmic steps, and select LR $= 5 \times 10^{-4}$ for all subsequent runs. The batch size is set to 256, and weight decay is 0 by default (except in weight-decay experiments). For the C-MNIST experiments we use LeNet-3 [26] with a 1024 embedding space, (as in Figure 1), we use the learning rate of LR $= 1 \times 10^{-5}$ and train for 50 epochs from random initialization.

For both COCO and ImageNet-1K we use TorchVision [35] augmentations. Training augmentations include (1) Random Resized Crop to a $224 \times 224$ patch (scaling and cropping with a random area and aspect ratio), (2) Color Jitter applied with probability $p = 0.8$ (brightness $\pm 40\%$, contrast $\pm 40\%$, saturation $\pm 20\%$, hue $\pm 10\%$), (3) Random Grayscale with $p = 0.2$, (4) Random Horizontal Flip with

Table 8: Hyperparameter tuning ranges per method and dataset.

| Method | Parameter | ImageNet-1K | COCO |
|---|---|---|---|
| Weight Decay | $\lambda_{\text{wd}}$ | $\{10^{-1}, 10^{-2}, \ldots, 10^{-6}\}$ | $\{10^{-1}, 10^{-2}, \ldots, 10^{-6}\}$ |
| L1 Sparse Activations | $\lambda_{\text{act}}$ | $\{10^{-1}, 10^{-2}, \ldots, 10^{-6}\}$ | $\{10^{-1}, 10^{-2}, \ldots, 10^{-6}\}$ |
| Group Lasso Activations | $\lambda_{\text{act}}$ | $\{10^{-1}, 10^{-2}, \ldots, 10^{-6}\}$ | $\{10^{-1}, 10^{-2}, \ldots, 10^{-6}\}$ |
| L1 Sparse Weights | $\lambda_{\text{weights}}$ | $\{10^{-1}, 10^{-2}, \ldots, 10^{-6}\}$ | $\{10^{-1}, 10^{-2}, \ldots, 10^{-6}\}$ |
| Group Lasso Weights | $\lambda_{\text{weights}}$ | $\{10^{-1}, 10^{-2}, \ldots, 10^{-6}\}$ | $\{10^{-1}, 10^{-2}, \ldots, 10^{-6}\}$ |
| HBaR | $\lambda_x$ | $\{0, 0.001, 0.005, 0.01\}$ | $\{0, 0.001, 0.005, 0.01\}$ |
| | $\lambda_y$ | $\{0, 0.005, 0.05, 0.5\}$ | $\{0, 0.001, 0.01, 0.1\}$ |
| | $\sigma$ | $\{0.5, 1.0, 5.0\}$ | $\{0.5, 5.0\}$ |
| H-SPLID | $\lambda_s$ | $\{0, 0.1, 0.5, 1.0\}$ | $\{0, 0.1, 0.5, 1.0\}$ |
| | $\lambda_n$ | $\{0, 0.05, 0.1\}$ | $\{0, 0.05, 0.1, 1.0\}$ |
| | $\rho_s$ | $\{0, 0.1, 0.15, 0.5\}$ | $\{0, 0.1, 0.15, 0.5\}$ |
| | $\rho_n$ | $\{0, 0.05, 0.1, 0.15, 0.2, 0.3\}$ | $\{0, 0.05, 0.1, 0.15, 0.2, 0.3\}$ |

Table 9: H-SPLID best hyperparameters for ImageNet-1K and COCO.

| Parameter | ImageNet-1K | COCO |
|---|---|---|
| $\beta_{\text{init\_fraction}}$ | 20% | 100% |
| $\beta_{\text{update\_fraction}}$ | 5% | 100% |
| $\beta_{\text{step}}$ | 0.995 | 0.8 |
| $\lambda_s$ | 0.1 | 0.1 |
| $\lambda_n$ | 0.1 | 0.2 |
| $\rho_s$ | 0.1 | 0.5 |
| $\rho_n$ | 0.1 | 0.05 |
| Shared space variation | 0.1 | 0.025 |

$p = 0.5$, (5) Random Solarize with threshold 0.5 and $p = 0.2$, followed by (6) `ToTensor` and (7) Normalization using per-channel means and standard deviations (ImageNet defaults $[0.485, 0.456, 0.406]$, $[0.229, 0.224, 0.225]$ or COCO-computed statistics). At test time, inputs are first resized so that the shorter side is 256 px, then center-cropped to $224 \times 224$, and finally passed through `ToTensor` and the same Normalization.

**Tuning strategy and per-method tuning ranges.** We employ identical hyperparameter tuning strategies for H-SPLID and all comparison methods. 30% of the training corpus is randomly sampled for ImageNet, while the complete training set is used for all other datasets. In either case, 20% of the samples are used to constitute the validation set. Hyperparameters are optimized via grid search by selecting the model configuration exhibiting the highest robust validation accuracy at the end of training, in which robustness is measured with respect to Projected Gradient Descent (PGD) [34] attacks applied to the entire image, so no knowledge of salient or non-salient regions is used. We use dataset-specific perturbation budgets of $\epsilon = \frac{1}{255}$ for ImageNet and $\epsilon = \frac{2}{255}$ for COCO. These values were chosen to be strong enough to select for more robust models, while at the same time being not too strong to induce model collapse to random accuracy, so we can use the metric for model selection. Based on this selection criterion we trained each method on COCO three times and selected the run with highest robust accuracy for validation. For ImageNet we only trained one run. Importantly, no information pertaining to the salient or non-salient regions is leveraged during the tuning. Table 8 summarizes the grid ranges we search for each method on ImageNet-1K and COCO.

**H-SPLID selected settings.** After tuning as above, the final hyperparameters chosen for H-SPLID on each dataset are listed in Table 9. In all H-SPLID runs we also set $\lambda_{ce} = 10$ to balance the cross-entropy scale. Due to the scale of ImageNet-1K, we introduce two scheduling parameters: (i) $\beta_{\text{init\_fraction}}$, the fraction of training data used to compute the initial mask values (20% for ImageNet-1K, 100% for COCO), and (ii) $\beta_{\text{update\_fraction}}$, which determines the amount of training data that must be processed before updating the masks (5% of the dataset for ImageNet-1K, corresponding to multiple updates per epoch; 100% for COCO, corresponding to one update per epoch).

Table 10: Best parameters per comparison method.

| Method | Parameter | ImageNet-1K | COCO |
|---|---|---|---|
| Weight Decay | $\lambda_{\text{wd}}$ | $10^{-5}$ | $10^{-3}$ |
| L1 Sparse Weights | $\lambda_{\text{weights}}$ | $10^{-6}$ | $10^{-5}$ |
| Group Lasso Weights | $\lambda_{\text{weights}}$ | $10^{-5}$ | $10^{-3}$ |
| L1 Sparse Activation | $\lambda_{\text{act}}$ | $10^{-4}$ | $10^{-6}$ |
| Group Lasso Activation | $\lambda_{\text{act}}$ | $10^{-4}$ | $10^{-5}$ |
| HBaR | $\lambda_x$ | $10^{-3}$ | $5^{-3}$ |
| | $\lambda_y$ | $10^{-1}$ | $5^{-2}$ |
| | $\sigma$ | 5 | 5 |

Table 11: Timing experiments on the COCO dataset.

| Method | average training time per epoch (in seconds) |
|---|---|
| Vanilla | 37.121 ($\pm$ 6.032) |
| Weight Decay | 37.687 ($\pm$ 6.680) |
| Group Lasso Weights | 37.353 ($\pm$ 6.151) |
| Group Lasso Activations | 37.354 ($\pm$ 6.455) |
| L1 Sparse Activations | 37.564 ($\pm$ 6.494) |
| L1 Sparse Weights | 37.450 ($\pm$ 6.217) |
| HBaR | 45.739 ($\pm$ 5.895) |
| H-SPLID | 42.852 ($\pm$ 9.672) |

Table 12: Runtime of ImageNet-1K experiments.

| Method | Runtime |
|---|---|
| Vanilla | 11h 27m |
| Weight Decay | 10h 22m |
| Group Lasso Activations | 10h 20m |
| Group Lasso Weights | 10h 25m |
| L1 Sparse Activations | 9h 53m |
| L1 Sparse Weights | 10h 30m |
| HBaR | 12h 04m |
| H-SPLID | 14h 6m |

**Baseline feature selection/regularisation methods.** Finally, Table 10 reports the single best regularization strength found for each of the comparison methods.

## G    Additional Experimental Results

### G.1    Training Time Comparison

**COCO Dataset.** The COCO experiments presented in the previous sections were executed for a total of 300 epochs on different machines with varying background workloads. To compare and report the computational intensity of the different training methods, the experiments were repeated on a single machine, albeit for a reduced number of epochs. Table 11 provides the average training time per epoch for the various training methods employed on the coco dataset. Those methods were executed for only 20 epochs on the same A100 GPU with 100 GB of memory, whence the average time per epoch was computed.

**ImageNet-1K Dataset.** The ImageNet-1K experiments were run for 20 epochs on an internal server (see Appendix F.3) that was not exclusively reserved for these trials, leading to varying background workloads. Logging was enabled throughout, with H-SPLID performing evaluations on the validation set every 5 epochs (during which additional metrics were recorded), while all other methods were evaluated every 10 epochs. These factors may contribute to H-SPLID's longer runtime. As a result,

Table 13: Sensitivity to $\lambda_s$.

| $\lambda_s$ | Clean Acc. (%) | PGD $\epsilon$=1/255 | PGD $\epsilon$=2/255 | PGD $\epsilon$=3/255 | Salient Dim. |
|---|---|---|---|---|---|
| 0.01 | 97.73 | 74.12±0.07 | 58.54±0.33 | 37.59±0.38 | 96 |
| 0.05 | 98.37 | 74.47±0.19 | 57.90±0.30 | 37.48±0.33 | 101 |
| 0.10 | 98.23 | 76.61±0.15 | 60.03±0.38 | 40.61±0.20 | 108 |
| 0.20 | 97.73 | 73.52±0.25 | 57.08±0.15 | 37.07±0.34 | 107 |
| 0.50 | 98.02 | 74.06±0.25 | 59.96±0.43 | 41.12±0.38 | 107 |
| 1.00 | 97.52 | 74.63±0.23 | 60.75±0.32 | 40.26±0.21 | 51 |

Table 14: Sensitivity to $\lambda_n$.

| $\lambda_n$ | Clean Acc. (%) | PGD $\epsilon$=1/255 | PGD $\epsilon$=2/255 | PGD $\epsilon$=3/255 | Salient Dim. |
|---|---|---|---|---|---|
| 0.01 | 97.80 | 72.66±0.17 | 55.02±0.29 | 35.83±0.48 | 199 |
| 0.05 | 98.23 | 76.61±0.15 | 60.03±0.38 | 40.61±0.20 | 108 |
| 0.10 | 98.09 | 76.40±0.09 | 61.06±0.42 | 40.52±0.30 | 29 |
| 0.20 | 97.45 | 79.02±0.11 | 68.01±0.33 | 57.07±0.10 | 14 |
| 0.50 | 61.30 | 50.40±0.03 | 43.95±0.32 | 35.07±0.15 | 5 |
| 1.00 | 32.74 | 32.74±0.00 | 32.74±0.00 | 32.74±0.00 | 0 |

the runtimes reported in Table 12 should be taken only as an overview and rough estimate rather than precise timing measurements. Due to the high computational cost of ImageNet-1K experiments and the lack of exclusive server access, precise timing experiments with exclusive access were conducted only on the COCO dataset (See Table 11).

### G.2 Hyperparameter Sensitivity on COCO-Animals

We conduct the sensitivity analysis starting from the best-performing configuration on COCO, varying $\lambda_s, \lambda_n, \rho_s, \rho_n, \beta_{\text{step}}$ while keeping all other settings fixed. In Tables 13–17, we report clean accuracy (%), PGD robustness at $\epsilon \in \{1, 2, 3\}/255$, and the learned salient dimensionality. Increasing $\lambda_n$ or $\rho_n$ generally improves robustness and compresses the salient subspace up to a regime where excessive regularization degrades performance. $\rho_s$ yields modest robustness gains with gradual salient-space shrinkage, and $\lambda_s$ exhibits a mild non-monotonic trend around the optimum. For $\beta_{\text{step}}$, very small values (e.g., 0.1) lead to model collapse, while intermediate values (0.3-0.5) reduce clean accuracy despite moderate robustness gains. Larger values (0.8-0.9) maintain high accuracy, with $\beta_{\text{step}} = 0.8$ achieving the best results. Overall, the best configuration attains a salient subspace of 14 (out of 512) dimensions with strong robustness.

Table 15: Sensitivity to $\rho_s$.

| $\rho_s$ | Clean Acc. (%) | PGD $\epsilon=1/255$ | PGD $\epsilon=2/255$ | PGD $\epsilon=3/255$ | Salient Dim. |
|------|------|------|------|------|------|
| 0.01 | 97.73 | 75.18±0.14 | 59.14±0.27 | 38.24±0.33 | 126 |
| 0.05 | 98.37 | 73.56±0.13 | 55.01±0.24 | 32.64±0.31 | 120 |
| 0.10 | 97.59 | 74.43±0.18 | 58.31±0.12 | 36.98±0.14 | 116 |
| 0.20 | 97.87 | 75.76±0.09 | 59.99±0.17 | 40.18±0.20 | 113 |
| 0.50 | 98.23 | 76.61±0.15 | 60.03±0.38 | 40.61±0.20 | 108 |
| 1.00 | 98.02 | 77.76±0.12 | 61.35±0.60 | 41.87±0.17 | 104 |

Table 16: Sensitivity to $\rho_n$.

| $\rho_n$ | Clean Acc. (%) | PGD $\epsilon=1/255$ | PGD $\epsilon=2/255$ | PGD $\epsilon=3/255$ | Salient Dim. |
|------|------|------|------|------|------|
| 0.01 | 98.09 | 74.59±0.17 | 57.07±0.21 | 37.76±0.29 | 124 |
| 0.05 | 98.23 | 76.61±0.15 | 60.03±0.38 | 40.61±0.20 | 108 |
| 0.10 | 97.52 | 76.48±0.16 | 59.80±0.30 | 39.33±0.16 | 86 |
| 0.20 | 97.52 | 75.41±0.11 | 57.76±0.14 | 37.93±0.44 | 57 |
| 0.50 | 32.74 | 32.74±0.00 | 32.74±0.00 | 32.74±0.00 | 0 |
| 1.00 | 32.74 | 32.74±0.00 | 32.74±0.00 | 32.74±0.00 | 0 |

Table 17: Sensitivity to $\beta_{\text{step}}$.

| $\beta_{\text{step}}$ | Clean Acc. (%) | PGD $\epsilon=1/255$ | PGD $\epsilon=2/255$ | PGD $\epsilon=3/255$ | Salient Dim. |
|------|------|------|------|------|------|
| 0.1 | 32.74 | 32.74±0.00 | 32.74±0.00 | 32.74±0.00 | 0 |
| 0.3 | 84.55 | 71.23±0.08 | 63.40±0.12 | 54.16±0.12 | 27 |
| 0.5 | 85.12 | 71.20±0.16 | 63.23±0.25 | 54.43±0.26 | 30 |
| 0.8 | 97.59 | 78.33±0.21 | 69.45±0.29 | 57.92±0.13 | 14 |
| 0.9 | 97.66 | 73.98±0.12 | 57.82±0.46 | 36.68±0.35 | 108 |
| 0.99 | 98.44 | 74.50±0.15 | 57.35±0.36 | 33.95±0.32 | 490 |

Table 18: Learning-rate study on COCO-Animals. Salient/Full is measured in a 512-d feature space.

| Method | Clean Acc. | PGD $\epsilon$=1/255 | PGD $\epsilon$=2/255 | PGD $\epsilon$=3/255 | AA $\epsilon$=1/255 | AA $\epsilon$=2/255 | AA $\epsilon$=3/255 | Salient/Full |
|---|---|---|---|---|---|---|---|---|
| H-SPLID LR=0.005 | 35.23 | 35.23±0.00 | 35.23±0.00 | 35.23±0.00 | 35.23±0.00 | 35.23±0.00 | 35.23±0.00 | 0 / 512 |
| H-SPLID LR=0.0005 | 97.87 | 80.01±0.15 | 69.44±0.16 | 56.95±0.34 | 75.18±0.08 | 58.64±0.17 | 50.40±0.19 | 14 / 512 |
| H-SPLID LR=0.00005 | 96.53 | 64.89±0.11 | 46.78±0.21 | 28.84±0.24 | 56.17±0.10 | 33.98±0.17 | 23.26±0.23 | 33 / 512 |
| H-SPLID LR=0.000005 | 92.49 | 54.30±0.03 | 35.07±0.23 | 17.53±0.20 | 46.82±0.07 | 21.59±0.16 | 11.92±0.11 | 450 / 512 |

## G.3 Effect of Learning Rate

To verify the effect of learning rate, we vary the learning rate (LR) around the best hyperparameter setting by powers of ten (see Table 18). Results indicate a strong influence on both robustness/accuracy and the learned subspace: $LR = 5 \times 10^{-4}$ yields the best overall performance with a compact salient subspace (14/512), lower LRs underfit and produce diffuse salient representations, and a higher LR collapses training.

