# OpenReview forum: "H-SPLID: HSIC-based Saliency Preserving Latent Information Decomposition"
_NeurIPS.cc/2025/Conference — NeurIPS 2025 poster_

### Official Review · Reviewer_HCiM · 2025-06-30

**Clarity:** 2
**Significance:** 2
**Originality:** 2
**Rating:** 4
**Confidence:** 3

**Summary:**

H-SPLID presents a method to decompose a network’s latent space into a low-dimensional salient subspace (task-relevant) and a non-salient subspace (task-irrelevant), using learnable binary masks and regularization via norm-based clustering losses and HSIC penalties. The authors prove (Theorem 3.2) that the expected prediction change under small input perturbations is upper-bounded by the HSIC between inputs and the salient representation, and the dimension of that subspace. Empirically, H-SPLID improves robustness to adversarial attacks targeted at backgrounds (COCO animals) and transfer performance on saliency benchmarks (ImageNet-9, CounterAnimal) without adversarial training

**Questions:**

1. More experiments on diverse datasets and modalities should be presented.
2. More studies about the parameters should be presented.

**Ethical Concerns:**

["NO or VERY MINOR ethics concerns only"]

**Final Justification:**

I have carefully read the comments from the other reviewers as well as the authors’ rebuttal. I believe the authors have addressed my concerns, and I concur with the other reviewers on the contributions of this work. Therefore, I am revising my rating to borderline accept.

**Limitations:**

See weakness and questions.

**Quality:**

3

**Strengths And Weaknesses:**

I am not an expert on this domain. I have some questions as follow
1. Experiments focus on image classification with fixed ResNets; no tests on other modalities (text, graphs) or real robotics pipelines. The synthetic MNIST concatenation and filtered COCO subsets do not capture the complexity of real-world feature entanglement.
2. The method introduces five new parameters (λₛ, λₙ, ρₛ, ρₙ, mask-update rates) and mask-update schedules (β-fractions, α-smoothing). Yet no study quantifies sensitivity or provides guidelines.
3. HSIC computation is O(n²) per minibatch—how does this scale to larger batch sizes or high-dimensional embeddings? Have the authors considered linearized/approximate HSIC estimators to reduce compute?

---

> ### Author Rebuttal · Authors · 2025-07-31
>
> # Response to Reviewer HCiM
>
> We thank the reviewer for their feedback and are happy to address all concerns raised below.
>
> ---
> ### **W1 and Q1:** Focus on image classification, inclusion of complex, real world data and additional modalities such as text
>
> Thank you for raising these important points. Since they concern multiple aspects of our work, we address each part individually below.
>
> **Focus on image classification.**
>
> Our main experiments are based on image classification tasks, as saliency concepts—such as foreground and background—are particularly well-defined and interpretable in computer vision. This makes it a natural domain for evaluating H-SPLID’s ability to disentangle salient and non-salient features. To this end, we tested H-SPLID on several challenging CV benchmarks.
>
> **Inclusion of complex, real-world data.**
>
> As shown in Table 2, we evaluate H-SPLID on datasets that explicitly model real-world feature entanglement. For instance, ImageNet9 [1] and its MixedRand variant (see Figure 4, Appendix F) introduce background shifts by replacing image backgrounds with those from other real-world classes, enabling evaluation under natural distribution changes.
>
> We also include the CounterAnimals dataset [2] (CA-Counter in Table 2), derived from iNaturalist wildlife images, which contrasts typical and atypical contexts (e.g., a polar bear on snow vs. grass; see Figure 3a). This setup provides a realistic test of generalization under spurious correlations.
>
> In addition, and in response to Reviewer 76kv (W1 & 2), we evaluated H-SPLID on the ISIC-2017 skin lesion classification dataset [3], comprising three real-world medical classes. H-SPLID consistently outperformed baselines under multiple real-world perturbations (see our response to Reviewer 76kv for details).
>
> **Inclusion of additional modality - text data**
>
> In contrast to saliency in images, modalities such as text involve more nuanced, task-specific definitions of saliency, where the notion of foreground versus background is less explicit. As such, further exploration is required to understand how H-SPLID's decomposition applies across modalities and how to appropriately define and measure saliency in those contexts. However, to illustrate the potential of H-SPLID beyond vision, we conducted an experiment on a text classification task using the ERASER Movie Reviews dataset [4]. In this benchmark, each example is annotated with human-provided rationales, which we treat as salient tokens and the task is to predict the reviews' sentiment (positive or negative).
>
> An example from ERASER movie reviews benchmark: "In this movie, **the acting is great!** The soundtrack is run-of-the-mill, **but the action more than makes up for it**." The highlighted texts are the rationales for the positive review. These are annotated by human experts. All other tokens we consider as non-salient (background).
>
> We adopted a new backbone—DistilBERT [5]—and evaluated robustness using the HotFlip adversarial attack [6], adapted to perturb *non-salient* tokens only. This setup tests whether the model relies on core (rationale) features or is vulnerable to noise in peripheral content.
>
> Based on the gradient, a HotFlip-1 attack would change one letter. Using the example above:
> *Attacked text:* "In this movie, the acting is great! The soun**T**track is run-of-the-mill, but the action more than makes up for it." --> Changing the sentiment prediction from positive to negative.
>
> Using limited hyperparameter tuning for H-SPLID, it improves adversarial robustness over the DistilBERT baseline, reducing the accuracy drop under both 1-token and 3-token perturbations:
>
> | Method            | Clean Accuracy | HotFlip-1 Accuracy | HotFlip-3 Accuracy | Max Accuracy Drop |
> |-------------------|----------------|---------------------|---------------------|--------------------|
> | DistilBERT      | 90.5%          | 76.4%               | 73.9%               | 16.6%              |
> | H-SPLID (Ours)     | 86.4%          | **82.9%**           | **75.4%**           | **11.1%**          |
>
> These results, while preliminary, suggest that H-SPLID can improve robustness even in textual domains. We acknowledge, however, that stability and generalization across modalities such as graphs and robotics pipelines remain open challenges. Arguably, an in-depth analysis and application of H-SPLID in domains such as text would be an independent contribution in its own right, perhaps even a separate paper.
>
> ---
> ### **W2:** Sensitivity study of introduced hyperparameters
>
> We conducted a detailed hyperparameter sensitivity analysis on the COCO-Animals dataset by varying one hyperparameter at a time, starting from the best-performing configuration. The tables below report the model’s performance in terms of Clean Accuracy (\%), PGD adversarial robustness across different attack strengths, and the dimensionality of the salient space. Each table shows how changes to a single hyperparameter affect these metrics while keeping all others fixed.
>
> | λₛ | Clean Acc. (%) | PGD ε=1/255 | PGD ε=2/255 | PGD ε=3/255 | Salient Dim. |
> | --- | --- | --- | --- | --- | --- |
> | 0.01 | 97.73 | 74.12±0.07 | 58.54±0.33 | 37.59±0.38 | 96 |
> | 0.05 | 98.37 | 74.47±0.19 | 57.90±0.30 | 37.48±0.33 | 101 |
> | 0.10 | 98.23 | 76.61±0.15 | 60.03±0.38 | 40.61±0.20 | 108 |
> | 0.20 | 97.73 | 73.52±0.25 | 57.08±0.15 | 37.07±0.34 | 107 |
> | 0.50 | 98.02 | 74.06±0.25 | 59.96±0.43 | 41.12±0.38 | 107 |
> | 1.00 | 97.52 | 74.63±0.23 | 60.75±0.32 | 40.26±0.21 | 51 |
>
> | λₙ | Clean Acc. (%) | PGD ε=1/255 | PGD ε=2/255 | PGD ε=3/255 | Salient Dim. |
> | --- | --- | --- | --- | --- | --- |
> | 0.01 | 97.80 | 72.66±0.17 | 55.02±0.29 | 35.83±0.48 | 199 |
> | 0.05 | 98.23 | 76.61±0.15 | 60.03±0.38 | 40.61±0.20 | 108 |
> | 0.10 | 98.09 | 76.40±0.09 | 61.06±0.42 | 40.52±0.30 | 29 |
> | 0.20 | 97.45 | 79.02±0.11 | 68.01±0.33 | 57.07±0.10 | 14 |
> | 0.50 | 61.30 | 50.40±0.03 | 43.95±0.32 | 35.07±0.15 | 5 |
> | 1.00 | 32.74| 32.74±0.00 | 32.74±0.00 | 32.74±0.00 | 0 |
>
> | ρₛ | Clean Acc. (%) | PGD ε=1/255 | PGD ε=2/255 | PGD ε=3/255 | Salient Dim. |
> | --- | --- | --- | --- | --- | --- |
> | 0.01 | 97.73 | 75.18±0.14 | 59.14±0.27 | 38.24±0.33 | 126 |
> | 0.05 | 98.37 | 73.56±0.13 | 55.01±0.24 | 32.64±0.31 | 120 |
> | 0.10 | 97.59 | 74.43±0.18 | 58.31±0.12 | 36.98±0.14 | 116 |
> | 0.20 | 97.87 | 75.76±0.09 | 59.99±0.17 | 40.18±0.20 | 113 |
> | 0.50 | 98.23 | 76.61±0.15 | 60.03±0.38 | 40.61±0.20 | 108 |
> | 1.00 | 98.02 | 77.76±0.12 | 61.35±0.60 | 41.87±0.17 | 104 |
>
> | ρₙ | Clean Acc. (%) | PGD ε=1/255 | PGD ε=2/255 | PGD ε=3/255 | Salient Dim. |
> | --- | --- | --- | --- | --- | --- |
> | 0.01 | 98.09 | 74.59±0.17 | 57.07±0.21 | 37.76±0.29 | 124 |
> | 0.05 | 98.23 | 76.61±0.15 | 60.03±0.38 | 40.61±0.20 | 108 |
> | 0.10 | 97.52 | 76.48±0.16 | 59.80±0.30 | 39.33±0.16 | 86 |
> | 0.20 | 97.52 | 75.41±0.11 | 57.76±0.14 | 37.93±0.44 | 57 |
> | 0.50 | 32.74 | 32.74±0.00 | 32.74±0.00 | 32.74±0.00 | 0 |
> | 1.00 | 32.74 | 32.74±0.00 | 32.74±0.00 | 32.74±0.00 | 0 |
>
> The hyperparameter sensitivity study on the COCO-Animals dataset highlights that $\lambda_n$ and $\rho_n$ are key to balancing adversarial robustness and the dimensionality of the salient space. Notably, the best model exhibits a salient dimensionality of just 14 (out of 512), indicating highly effective compression of salient features.
>
> In general, increasing $\lambda_n$ and $\rho_n$ improved robustness and reduced the salient dimension, although very high values led to performance collapse. For $\lambda_s$, robustness initially improved before slightly declining, while increasing $\rho_s$ yielded modest but consistent gains in robustness—both also contributing to a reduction in salient dimensionality. We will include the remaining analysis on mask update rates and scheduling strategies in the camera-ready version of the paper.
>
> ### **W3:** Scaling of HSIC computation to high-dimensional embeddings
>
>
> **Empirical scalability.** Our method is practically feasible, as demonstrated by competitive training times on large-scale datasets such as ImageNet-1K with a ResNet50 backbone (embedding dimension 2048), as shown in Tables 9 and 10. This suggests that H-SPLID is viable for real-world applications. Moreover, further approximations can extend its scalability while preserving statistical reliability and interpretability.
>
> **Approximation strategies.** While we adopt the HSIC implementation from HBaR [7], various approximation techniques can further reduce its cost. Block-based HSIC [8], for example, partitions each minibatch into blocks of size $B$, computes HSIC within each block, and averages the results. This reduces complexity to $\mathcal{O}(Bn)$, making it suitable for large batches. In addition, general kernel approximation methods such as random Fourier features and Nyström approximations [8,9] offer promising directions for future extensions.
>
>
> ---
>
> Overall we want to thank the reviewer for the detailed feedback, it helped to further improve our paper.
>
> ---
>
>
> ### References
> [1] Xiao, et al: Noise or Signal: The Role of Image Backgrounds in Object Recognition. ICLR 2021
>
> [2] Wang, et al. A Sober Look at the Robustness of CLIPs to Spurious Features. NeurIPS 2024
>
> [3] Codella, et al. Skin Lesion Analysis Toward Melanoma Detection: A Challenge at the 2017 International Symposium on Biomedical Imaging (ISBI). arXiv: 1710.05006
>
> [4] DeYoung, et al. ERASER: A Benchmark to Evaluate Rationalized NLP Models. ACL, 2020.
>
> [5] Sanh, et al. DistilBERT, a Distilled Version of BERT: Smaller, Faster, Cheaper and Lighter. *arXiv preprint* arXiv:1910.01108, 2019.
>
> [6] Ebrahimi, et al. HotFlip: White-Box Adversarial Examples for Text Classification. ACL, 2018.
>
> [7] Wang, et al. Revisiting hilbert-schmidt information bottleneck for adversarial robustness. NeurIPS 2021.
>
> [8] Zhang,et al. Large-scale kernel methods for independence testing. Statistics and Computing, 28(1), 2018.
>
> [9] Ninh and Pagh. Fast and scalable polynomial kernels via explicit feature maps. KDD 2013.

---

> > ### Author Response · Authors · 2025-08-05
> > **Official Comment by Authors**
> >
> > Dear Reviewer,
> >
> > Thank you for your time and thoughtful feedback. We have provided extensive answers to your review and hope that our responses have addressed your concerns.
> >
> > With three days remaining in the discussion period, please let us know if you have any further questions. If our responses were satisfactory, we would be grateful if you would consider updating your score.

---

> > ### Comment · Reviewer_HCiM · 2025-08-06
> >
> > I have carefully read the comments from the other reviewers as well as the authors’ rebuttal. I believe the authors have addressed my concerns, and I concur with the other reviewers on the contributions of this work. Therefore, I am revising my rating to borderline accept.

---

> > > ### Author Response · Authors · 2025-08-06
> > > **Official Comment by Authors**
> > >
> > > Thank you for taking the time to read our rebuttal and the comments from the other reviewers. We're pleased that our additional experiments and clarifications helped address your concerns. Your feedback prompted meaningful improvements to the paper, and we sincerely appreciate the time and care you dedicated to the review process.

---

### Official Review · Reviewer_76kv · 2025-07-01

**Clarity:** 3
**Significance:** 3
**Originality:** 3
**Rating:** 5
**Confidence:** 4

**Summary:**

This paper introduces H-SPLID, an algorithm for learning salient feature representations through explicit decomposition of latent spaces. The H-SPLID approach leverages information compression regularization and the Hilbert-Schmidt Independence Criterion to promote compact and informative representations within a task-relevant subspace. Some experimental results are provided which demonstrate H-SPLID’s efficacy in mitigating the vulnerability to irrelevant input variations, linking representation dimensionality and robustness.

**Questions:**

- Does the learning rate influence the final outcome, e.g. for the salience subspace versus the non-salient subspace?
- Why PGD over any other attack?

**Ethical Concerns:**

["NO or VERY MINOR ethics concerns only"]

**Final Justification:**

In final evaluation and in light of the author rebuttal responses (including thorough analysis and extended results, which they promise to include in the revision), I update my recommended score to reflect a better balanced paper with clear contributions.

**Limitations:**

yes

**Quality:**

3

**Strengths And Weaknesses:**

Strengths:
+ Figure 1 is compelling and informative, demonstrating the strengths of the proposed approach in a visual manner with a clear t-SNE visualization.
+ The problem is of significance to the community - the ability to learn representations robust to irrelevant input is a foundational requirement for deploying deep neural networks in real-world scenarios.

Weaknesses:
- Relying heavily on adversarial attacks to demonstrate robustness is a somewhat narrow approach. While it’s effective to some extent, exploring the method's performance under more natural, real-world perturbations would be more valuable. (Suggestions: lighting, blur, occlusion, etc)
- There is a limitation in the experimentation on COCO and ImageNet datasets - extended analysis beyond these and ResNet models would make for a more compelling demonstration of the capabilities of the proposed H-SPLID. Even evaluations on a medical imaging dataset for example would convey better the breadth of applicability of the approach as well as robustness to “real world” corruptions.

---

> ### Author Rebuttal · Authors · 2025-07-31
>
> # Response to Reviewer to 76kv
>
> Thank you for your thoughtful feedback. We appreciate the recognition of the problem’s significance—learning representations robust to irrelevant input is indeed essential for real-world deployment of deep neural networks. We address the open questions and concerns below.
>
> ---
>
> ### **W1 and W2:** Additional results on medical imaging dataset and real world perturbations
>
> We would like to clarify that while we do explore the effect of adversarial attacks on the COCO dataset (Table 1), our evaluation already incorporates datasets offering additional perspectives on robustness. For instance, the ImageNet9 [1] dataset (see Figure 4) introduces background modifications, with its MixedRand version occluding the background of the ImageNet images with background taken from other natural images (see Figure 4 in Appendix F for an example). Furthermore, we already include experiments with natural, real-world perturbations using the Counter animals [2] dataset (abbreviated as CA-Counter in Table 2). The counter animals dataset is derived from iNaturalist wildlife photos and is split into a common subset (featuring typical backgrounds) and a counter subset (with atypical yet plausible backgrounds). For example, as shown in Figure 3a, a common background for an ice bear is a winter landscape, whereas an atypical background for an ice bear would be green grass.
>
> In response to Reviewer HCiM’s suggestion to include a text-based dataset, we further evaluated H-SPLID on the ERASER Movie Reviews dataset [3], a benchmark for sentiment analysis that provides human annotated rationales.
>
> We fully agree that the inclusion of a **medical imaging dataset** would further strengthen the paper's demonstration of broader applicability and robustness. To this end, we have conducted new experiments on the ISIC-2017 skin lesion classification dataset [4], which comprises three classes: nevus, melanoma, and seborrheic keratosis. For this, we used a ResNet-50 pre-trained on ImageNet for feature extraction and trained a three-class classification head for 50 epochs, followed by 50 epochs of method-specific training. Model tuning and selection were conducted based on the procedure outlined in Appendix F.4.
>
> To evaluate robustness against the reviewer-suggested **real-world perturbations**: lighting, blur, and occlusion - we utilized the implementations of brightness, defocus blur, and occlusions$^1$ from the corruptions benchmark [5]. Following the analysis in the paper, we perturbed only the non-salient regions, such as non-lesion pixels. The results averaged over 10 random seeds are presented below:
>
>
> | Model | Clean Acc | Brightness | Defocus Blur | Occlusion|
> |---|---|---|---|---|
> | **H-SPLID** | **76.78±0.778** | **70.00±1.619** | **68.38±1.376** | **69.50±1.716** |
> | **Group Lasso Activations** | 75.38±1.383 | 66.50±3.136 | 61.32±4.501 | 62.68±2.969 |
> | **Group Lasso Weights** | 70.65±4.118 | 60.23±6.698 | 58.63±9.564 | 60.62±5.697 |
> | **HBaR** | **75.90±0.844** | **68.70±1.942** | 65.62±2.058 | 66.18±3.013 |
> | **L1 Sparse Activations** | 75.62±1.211 | 66.50±2.171 | 64.33±1.653 | 62.27±4.256 |
> | **L1 Sparse Weights** | 75.30±1.040 | 66.13±2.432 | 63.22±2.506 | 62.70±3.810 |
> | **Vanilla** | 75.45±0.986 | 66.43±2.527 | 63.77±2.388 | 62.87±3.081 |
> | **Weight Decay** | 75.63±1.545 | 67.53±2.980 | 64.57±2.295 | 63.55±3.851 |
>
> Our method, H-SPLID, outperforms the competitors across all three perturbation types on the ISIC-2017 dataset, further demonstrating its robustness to more natural, real-world perturbations and its applicability in specialized domains like medical imaging. We thank the reviewer for the suggestion and we will incorporate the medical imaging dataset analysis into the revised paper.
>
>
> ----
>
> $^1$ The snow perturbation is effectively occluding small patches of the image with white snow, thus we refer to it as occlusion. An example is given in Figure 1 of the original paper [5].
>
> ---
>
> ### **Q1:** Does the learning rate influence the final outcome, e.g. salience subspace versus the non-salient subspace?
>
> Yes, the learning rate is an important hyperparameter for deep learning methods in general, and our H-SPLID approach is no exception to that. We conducted additional experiments on the COCO-Animals dataset w.r.t. the learning rate, where we take the model with the best hyperparameters and vary the learning rate by powers of ten to test the learning rates influence.
>
> | Method                   | Clean Acc     | PGD ε=1/255 | PGD ε=2/255 | PGD ε=3/255 | AA ε=1/255 | AA ε=2/255 | AA ε=3/255 | Salient Dim. / Full Dim. |
> |--------------------------|---------------|-------------|-------------|-------------|-------------|-------------|-------------|----------------------------|
> | **H-SPLID LR=0.005**     | 35.23 | 35.23±0.00 | 35.23±0.00 | 35.23±0.00 | 35.23±0.00 | 35.23±0.00 | 35.23±0.00 | 0 / 512 |
> | **H-SPLID LR=0.0005**    | 97.87 | 80.01±0.15 | 69.44±0.16 | 56.95±0.34 | 75.18±0.08 | 58.64±0.17 | 50.40±0.19 | 14 / 512 |
> | **H-SPLID LR=0.00005**   | 96.53 | 64.89±0.11 | 46.78±0.21 | 28.84±0.24 | 56.17±0.10 | 33.98±0.17 | 23.26±0.23 | 33 / 512 |
> | **H-SPLID LR=0.000005**  | 92.49 | 54.30±0.03 | 35.07±0.23 | 17.53±0.20 | 46.82±0.07 | 21.59±0.16 | 11.92±0.11 | 450 / 512 |
>
> Our experiments demonstrate that the learning rate significantly influences both: the model’s performance, as well as the learned salient dimension. Specifically, a learning rate of 0.0005 yielded the best overall performance, achieving the highest clean accuracy and robustness across different strengths of PGD and AutoAttacks (AA). At this learning rate, the salient dimension was found to be 14 out of 512. Lower learning rates (e.g., 0.000005 and 0.00005) resulted in progressively lower accuracy and robustness, indicating that the model did not learn an informative salient representation. Notably, a learning rate of 0.000005 resulted in a significantly higher salient dimension (450/512), suggesting that a very low learning rate might lead to a less compact salient space. Conversely, a higher learning rate (0.005) led to a complete collapse of performance, with the model failing to learn any meaningful representation (salient dimension of 0/512). We will include these results in to the paper.
>
> ---
>
> ### **Q2:** Why PGD over any other attack?
>
> We would like to draw the reviewer's attention to Table 1, where we not only compare robustness w.r.t. PGD, but also to AutoAttack. AutoAttack serves as a strong attack that uses an ensemble of diverse parameter-free adversarial attacks.
>
>
> ---
> We thank the reviewer for the extensive feedback and will revise the final version by including the suggested experiments and clarifications. These will further strengthen our work.
>
> ---
>
>
> ### References
>
> [1] Kai Yuanqing Xiao, Logan Engstrom, Andrew Ilyas, Aleksander Madry: Noise or Signal: The Role of Image Backgrounds in Object Recognition. ICLR 2021
>
> [2] Qizhou Wang, Yong Lin, Yongqiang Chen, Ludwig Schmidt, Bo Han, Tong Zhang: A Sober Look at the Robustness of CLIPs to Spurious Features. NeurIPS 2024
>
> [3] DeYoung, Jay, et al. "ERASER: A Benchmark to Evaluate Rationalized NLP Models." *Transactions of the Association for Computational Linguistics*, 2020.
>
> [4] Codella N, Gutman D, Celebi ME, Helba B, Marchetti MA, Dusza S, Kalloo A, Liopyris K, Mishra N, Kittler H, Halpern A. "Skin Lesion Analysis Toward Melanoma Detection: A Challenge at the 2017 International Symposium on Biomedical Imaging (ISBI), Hosted by the International Skin Imaging Collaboration (ISIC)". arXiv: 1710.05006
>
> [5] Dan Hendrycks and Thomas Dietterich: Benchmarking Neural Network Robustness to Common Corruptions and Perturbations. ICLR 2019.

---

> > ### Author Response · Authors · 2025-08-05
> > **Official Comment by Authors**
> >
> > Dear Reviewer,
> >
> > Thank you for your valuable feedback. Your comments have further improved the quality of our work and strengthened our results.
> >
> > We are writing to inquire whether you have any further questions we could clarify during the discussion period (which ends in 3 days). We hope we have addressed your concerns. If so, we kindly ask you to consider updating your score.

---

> > > ### Comment · Reviewer_76kv · 2025-08-05
> > > **Response**
> > >
> > > I see that my response had not posted.
> > > Thank you for your detailed responses and the inclusion of extensive additional support to the themes highlighted across the reviews.
> > > In response to Q1 (beyond the rhetorical statements on LR importance) the analysis of its effect upon the proposed approach is helpful and should be included in the discussion.
> > > While Q2 is acknowledged, the question was not whether additional attacks were included but why they were chosen. It is still my recommendation that a (very brief) indication as to the choice of attacks is made clear in the text.
> > > Given the further experiments and analysis, with clear results, I am ready to change my review at this time. Thank you.

---

> > > > ### Author Response · Authors · 2025-08-05
> > > >
> > > > Thank you for taking the time to read our rebuttal and for your thoughtful follow‑up. We’re really glad to hear that the additional experiments, analyses, and clarifications were helpful. We’ll make sure to include the discussion on learning rate effects and add a brief explanation for our choice of attacks, as you suggested. Your constructive feedback has been invaluable in strengthening the paper, and we truly appreciate the time and care you’ve put into your feedback.

---

### Official Review · Reviewer_XsTp · 2025-07-01

**Clarity:** 3
**Significance:** 3
**Originality:** 3
**Rating:** 6
**Confidence:** 3

**Summary:**

This paper aims to learn salient features for supervised learning tasks, separating out non-salient features into a subset of the feature space dimensions. Unlike previous methods, the proposed H-SPLID algorithm learns to separate these salient and non-salient features without additional data that goes beyond the data used for supervised learning. The loss function contains 3 types of terms. The cross entropy term is the cross entropy loss when predicting from only the salient features. The masked clustering terms encourage features to match mean features, with class-specific means for salient features and a global mean for non-salient features. Lastly, the HSIC terms minimize the dependence between the data and the salient features as well as the labels and the non-salient features. The H-SPLID algorithm minimizes the loss function through alternating optimization of the neural network and the saliency mask. H-SPLID is supported by a theoretical result that bounds the change in model prediction by the HSIC of the data and the salient features. Experiments show that the method improves robustness to adversarial attacks and transfer accuracy for saliency benchmarks. An ablation additionally shows that performance degrades when only a subset of the loss terms are used.

**Questions:**

Q1: I think the end of line 115 should use the identity matrix as in line 114.

Q2: Is there a difference between $L_{ST}$ and Equation 5? It’s ambiguous why the different notation for the loss differs from the left side of Equation 5. Did you introduce the additional notation for when a minibatch is used instead of the full dataset? Are $L_s$ and $L_n$ used in both steps of H-SPLID or just the mask update?

Q3: The equation after line 152 made me confused about the definition of $\beta$. My initial assumption was that $\beta$ was a vector of 0 or 1 values. But instead line 113 says that $\beta$ is exactly one 0 or 1 value, which doesn’t make sense in the context of the mask definition in the previous line. Line 112 seems to best correlate with the equation after line 152, which seems to imply that each of the elements of $\beta$ lie in the range $[0, 1]$, although that definition is a bit overly broad since it doesn’t capture the diagonal nature of the mask matrix.

Q4: What variable is the expectation in Equation 9 taken with respect to? Looking at Appendix C suggests to me that it is with respect to $\delta$, but I would like that to be more explicitly stated.

**Ethical Concerns:**

["NO or VERY MINOR ethics concerns only"]

**Final Justification:**

The authors have very eloquently resolved the clarity issues I raised with the paper, so I have raised my recommendation to a strong accept.

**Limitations:**

Yes.

**Quality:**

3

**Strengths And Weaknesses:**

S1: The design of the experiments is well documented, including a detailed description of the datasets, methods, experiment process, and performance metrics. It seems like the experiment could be reproduced even without the code that is additionally provided.

S2: The proof in Appendix C appears to be a significant improvement over previous work and represent a lot of effort on the authors’ part. Although I am not familiar enough with some of the mathematical concepts to give a rigorous evaluation of the proof, I do not see any errors. The outline of the proof provided by some of the guiding text was very useful for understanding the logic of the proof.

W1: The mathematical notation has various issues throughout the paper (Q1-4). These issues reduced clarity and necessitated educated guesses as to some specifics of the method.

W2: It’s unclear to me how much benefit there is in learning non-salient features. Are these features expected to frequently be turned into salient features throughout training? As I understand it, the non-salient features do not affect the final prediction.

---

> ### Author Rebuttal · Authors · 2025-07-31
>
> # Response to Reviewer XsTp
>
> We thank the reviewer for their thoughtful and detailed assessment of our work. We are particularly grateful for their recognition of the clarity and reproducibility of our experimental design, and the effort invested in the theoretical analysis. We also appreciate the positive remarks on the structure and accessibility of the proof provided in Appendix C. We address the remaining questions and concerns below.
>
> ---
>
> ### **W2:** Clarification on the role and benefit of learning non-salient features
>
> Thank you for the thoughtful question. While non-salient features do not directly affect the final prediction, they play an important role during training, as salient features are defined in contrast to them. Their presence is therefore essential. By capturing task-irrelevant variations—such as background noise—non-salient features help isolate predictive factors in the salient space, enhancing robustness and disentanglement. These features are not expected to become salient over time; rather, the model learns a stable partition of latent dimensions based on the H-SPLID loss. We will clarify this in the paper.
>
> More broadly, our approach serves as a novel form of regularization. Although deep neural networks are highly expressive, their performance can degrade due to entanglement with irrelevant features. By jointly learning the classification task and the decomposition into salient and non-salient components, our method improves robustness, as shown in our experiments. In essence, H-SPLID encourages the network to focus on the most relevant information during training.
>
> ---
>
> ### **Q1:** Typo in line 115 (identity matrix)
>
> Thank you for pointing out the typo. We will fix it.
>
> ---
>
> ### **Q2a:**  Difference between $L_{ST}$ and Equation 5?
>
> We apologize for the typo and confusion. They are the same thing. We will unify the notation.
>
> ### **Q2b:** Are $L_s$ and $L_n$ used in both steps of H-SPLID or just the mask update?
>
> $L_s$ and $L_n$ are used in both steps. As they will allow backpropagation to update the encoder $f_\psi$.
>
> ---
>
> ### **Q3:** Definition of β and range of its values
>
> Thank you for pointing out this ambiguity—we apologize for the omission in the original text. To clarify, we apply a hard threshold to the continuous solution for $\beta$, mapping it to $\lbrace 0, 1 \rbrace^d$. While the optimization initially produces a continuous vector $\beta \in [0,1]^d$, we threshold each component at 0.5 to obtain a binary mask. We first experimented with using the continuous solution directly (i.e., without thresholding), and found that the learned values typically concentrate near 0 or 1. As a result, both variants—the soft (continuous) and hard (thresholded) masks—perform similarly in practice. We will revise the manuscript to explicitly define $\beta$ as a vector in $\lbrace 0, 1 \rbrace^d$, clarifying its role as the diagonal of the mask matrix, and to explain the thresholding procedure it entails.
>
> ---
>
> ### **Q4:** What variable is the expectation in Equation 9 taken with respect to?
>
> The expectation in Equation (9) is taken with respect to $x$, i.e.:
>
> $\mathbb{E}_x\lbrack \cdot \rbrack$,
>
> where $x$ is sampled from the data distribution. The perturbation $\delta$ is an arbitrary vector constrained in norm by $r$, and not a random variable. This distinction is explicitly stated in the theorem and elaborated in Appendix C, but we acknowledge that the notation in Equation (9) could have been clearer. To avoid any confusion, we will revise the equation to use $\mathbb{E}_{x}$ explicitly in the camera-ready version.
>
> ---
>
> We appreciate the reviewer’s careful reading and the opportunity to clarify the content of our paper. We have already incorporated the reviewer's suggestion into our manuscript.

---

> > ### Author Response · Authors · 2025-08-05
> > **Official Comment by Authors**
> >
> > Dear Reviewer,
> >
> > Thank you for your thoughtful and detailed assessment. We appreciate your positive feedback on the clarity and reproducibility of our experiments, as well as the effort invested in our theoretical proof. We appreciate your careful reading and the opportunity to clarify our work. We are writing to inquire whether you have any further questions we could clarify during the discussion period (which ends in 3 days).

---

### Official Review · Reviewer_P27a · 2025-07-02

**Clarity:** 3
**Significance:** 3
**Originality:** 3
**Rating:** 5
**Confidence:** 2

**Summary:**

This paper presents H‑SPLID, an algorithm that explicitly separates salient from non‑salient features by learning two disjoint latent sub‑spaces. The authors demonstrate its effectiveness on two image‑classification benchmarks and under two adversarial attack settings.

**Questions:**

1. What was the selection criteria for the COCO subset?
2. Why didn't the authors provide more experiments on several classifiers?

**Ethical Concerns:**

["NO or VERY MINOR ethics concerns only"]

**Final Justification:**

Rebuttal responses clarified my doubts, I consider this paper valuable

**Limitations:**

yes

**Quality:**

3

**Strengths And Weaknesses:**

Strengths:
- The paper is technically sound and its claims are supported by a thorough theoretical analysis in both the main text and the appendix.
- The empirical evaluation is comprehensive and well explained.
- The exposition is generally clear, and the authors provide code and full hyperparameter details, ensuring reproducibility.
- Explicitly disentangling salient and non‑salient latent dimensions is a novel contribution that remains underexplored in existing work.

Weaknesses:
- Lack of tests with alternative backbones (e.g., ViT, Inception, VGG).
- Parts of the methodology appear unnecessarily intricate and could be streamlined (e.g. Sec. 3.1)

If the approach works on a broader set of architectures and tasks, it could meaningfully influence saliency‑based robustness research.

---

> ### Author Rebuttal · Authors · 2025-07-31
>
> # Response to Reviewer P27a
>
> We appreciate the reviewer’s positive feedback on our work, particularly their recognition of the novelty of our method and the thoroughness of our empirical and theoretical analysis. We address all remaining concerns and questions below.
>
> ---
>
> ### **W2:** Streamlining of methodology section (e.g. Sec. 3.1)
>
> We appreciate the suggestion to improve the clarity of the methodology section. In the camera-ready version, we will streamline the presentation of Section 3 by reorganizing the conceptual flow and presenting the loss terms in a more concise and intuitive manner. We will also polish the overall writing and notation to improve readability and precision.
>
> ---
>
> ### **Q1:** Selection criteria for the COCO subset
>
> We selected a COCO subset in which the object of interest is correlated with its background. We focused on images of animals in their natural habitats—e.g., giraffes in the savannah—where background and object are contextually linked. We also chose non-overlapping classes (e.g., excluding dogs and humans, which frequently co-occur) and prioritized images where the background occupies a large portion of the image, like for images of animals in the wild, to have enough room for attacks on the background. This setup provides a challenging test-bed for our approach, which learns salient object-specific features and non-salient background features. We will clarify this selection criterion in the paper.
>
> ---
>
> ###  **W1 and Q2:** Additional experiments on different classifiers (alternative backbones)
>
> We would like to clarify that our experiments already incorporate a variety of backbone architectures. Specifically, we use two ResNet variants of different capacities—ResNet18 for COCO and ResNet50 for ImageNet—as well as LeNet-3 [1] for C-MNIST (see Figure 1 and Appendix *F.4 Hyperparameters*).
>
> In response to Reviewer HCiM’s suggestion to include a text-based dataset, we further evaluated H-SPLID on the ERASER Movie Reviews dataset [2], a benchmark with human-annotated rationales.
>
> We adopted the DistilBERT language model [3] as a new backbone and assessed robustness using the HotFlip adversarial attack [4], which we adapted to adversarially perturb $l$ letters of non-salient "background" words. Here, we define the salient words by the rationale masks provided by human annotators. For example, given the text: "In this movie, **the acting is great!** The soundtrack is run-of-the-mill, **but the action more than makes up for it**.", the highlighted part was the rationale by a human annotator and the remaining words serve as "background" words that will be attacked.
>
>
> Thus, the sentiment analysis setup probes whether the model relies primarily on core (rationale) features or is sensitive to noise in peripheral content.
>
> | Method            | Clean Accuracy | HotFlip-1 Accuracy | HotFlip-3 Accuracy | Max Accuracy Drop |
> |-------------------|----------------|---------------------|---------------------|--------------------|
> | DistilBERT with weight decay       |**90.5%**          | 76.4%               | 73.9%               | -16.6%              |
> | H-SPLID (Ours)     | 86.4%          | **82.9%**           | **75.4%**           | **-11.1%**          |
>
> For HotFlip attacks with $l=1$ and $l=3$ H-SPLID outperforms the DistilBERT baseline. These preliminary results suggest that H-SPLID can transfer to different backbones and even to different modalities, such as text. Based on this result we see the in-depth study of H-SPLID across modalities as promising future work.
>
> ---
>
> We thank the reviewer for their thoughtful feedback and will revise the final version to further improve the presentation and highlight the generality of H-SPLID.
>
> ---
>
>
> ### References
>
> [1] Y. LeCun, B. Boser, J. S. Denker, D. Henderson, R. E. Howard, W. Hubbard, and L. D. Jackel.381 Backpropagation applied to handwritten zip code recognition. Neural computation, 1(4):382 541–551, 1989.
>
> [2] DeYoung, Jay, et al. "ERASER: A Benchmark to Evaluate Rationalized NLP Models." *Transactions of the Association for Computational Linguistics*, 2020.
>
> [3] Sanh, Victor, et al. "DistilBERT, a Distilled Version of BERT: Smaller, Faster, Cheaper and Lighter." *arXiv preprint* arXiv:1910.01108, 2019.
>
> [4] Ebrahimi, Javid, et al. "HotFlip: White-Box Adversarial Examples for Text Classification." *Proceedings of the 56th Annual Meeting of the Association for Computational Linguistics*, 2018.

---

> > ### Comment · Reviewer_P27a · 2025-08-04
> >
> > Thank you the authors for their response. After carefully reading the rebuttal and the other reviews, I still have some concerns that remain only partially addressed.
> >
> > Regarding Q1, the selection of the COCO subset, as described by the authors, may introduce selection bias, which raises questions about the generalizability of the proposed method to real-world datasets where class boundaries are more ambiguous. Additionally, while the authors correctly point out (in response to other reviewers) that they tested the method on real-world data such as CounterAnimal, it is also true that all the natural datasets involved in the work are limited to animal-related categories, which may not fully capture the diversity and complexity of broader real-world settings.
> >
> > For Q2, I appreciate the authors’ clarification and the inclusion of additional experiments in the text domain using transformer-based models. However, my original question referred specifically to experiments with different backbone architectures in the image domain, including transformer-based vision models (e.g., ViT). This concern remains unaddressed, despite the relevance of such architectures in computer vision.
> >
> > In summary, I still find the paper interesting and relevant, and I consider the proposed disentangling strategy valuable for enhancing the robustness and interpretability of learned representations. However, some of my concerns remain unresolved, and I will take into account them, the broader discussion and feedback from other reviewers also for my final rating.

---

> > > ### Author Response · Authors · 2025-08-04
> > > **Official Comment by Authors**
> > >
> > > We thank the reviewer for finding our paper interesting and relevant. We address the remaining concerns below:
> > >
> > > ---
> > >
> > > **Q1:** We would like to clarify that our evaluation goes beyond animal-related categories. In addition to the CounterAnimal and COCO subset, we also report results on the ImageNet-9 benchmark (Table 2 datasets IN-9, Only-FG, MixedRand), which includes a broader range of categories -- such as vehicles (e.g., cars, trucks, trains), musical instruments (e.g., guitars, drums, violins), and various objects (e.g., tools, equipment, etc.) -- thereby extending beyond the domain of animals. Furthermore, in response to Reviewer 76kv, we incorporated experiments on the ISIC-2017 medical imaging dataset for skin lesion classification. This inclusion further highlights the applicability of our method to complex real-world domains beyond natural scenes. Our preliminary results on a text classification task (ERASER Movie Reviews with HotFlip attacks) show that H-SPLID generalizes to other domains. Taken together, these experiments support the generalizability of our method and demonstrate its effectiveness beyond narrow or animal-related datasets.
> > >
> > > ---
> > >
> > > **Q2:** We acknowledge the reviewer’s concern about architectural diversity and provide further clarification. As outlined in the paper and our prior response, we have already evaluated H-SPLID on a range of architectures: ResNet-18 (for COCO), ResNet-50 (for ImageNet-9 and transfer experiments), and LeNet-3 (for C-MNIST), covering both shallow and deep convolutional backbones. Additionally, our text classification experiment on the ERASER Movie Reviews dataset uses DistilBERT, a transformer-based model – marking both a departure from convolutional architectures and an application to a different data modality. These results highlight the model-agnostic nature of H-SPLID: it operates directly on latent representations and imposes no architectural constraints, making it applicable to a wide variety of models, including transformers.

---

> > > > ### Comment · Reviewer_P27a · 2025-08-08
> > > >
> > > > I thank the authors for their response, after reading it and the responses of the other reviewers I will keep my score

---

### Note · Authors · 2025-08-11

Dear AC and Reviewers,

We sincerely thank you for your time, effort, and constructive feedback during the review process. We greatly appreciate the recognition of H-SPLID's significance and relevance to the community [P27a, 76kv], technical soundness [P27a, XsTp], and the thoroughness of our theoretical and empirical analyses [P27a, 76kv, XsTp]. Your comments have helped us further strengthen and clarify the work.

In response to your suggestions, we have conducted several additional experiments and analyses:

- **Broader domain evaluation** [76kv, HCiM]: Following the recommendation to test on medical data [76kv], we evaluated H-SPLID on the ISIC-2017 skin lesion classification dataset. Our method outperformed the competitors under clean conditions and under realistic perturbations such as lighting changes, blur, and occlusion, demonstrating robustness beyond adversarial attacks and applicability to specialized domains.

- **Architectural and modality diversity** [P27a, HCiM]: We clarified that our original experiments already include varied convolutional backbones (ResNet-18, ResNet-50, LeNet-3). Additionally, following HCiM's recommendation, we extended H-SPLID to the text domain, evaluating on the ERASER Movie Reviews benchmark with DistilBERT, a transformer-based backbone, under HotFlip adversarial perturbations. This preliminary study indicates that H-SPLID learns a salient representation even in non-vision modalities.

- **Hyperparameter and learning rate analysis** [HCiM, 76kv]: We performed a detailed sensitivity study on λₛ, λₙ, ρₛ, and ρₙ, as well as the learning rate. Our results show how these parameters influence both robustness and the compactness of the salient subspace, providing practical guidance for tuning H-SPLID.

- **Scalability considerations** [HCiM]: We discussed the computational complexity of HSIC, confirmed its practical feasibility for large-scale settings, and outlined possible approximation strategies for even larger models or datasets.

- **Theoretical proof** [XsTp]: We thank Reviewer XsTp for the effort invested in reviewing our theoretical proof. Their comments helped us streamline the methodology section, improve the clarity of the mathematical notation, and add explanations where ambiguity was noted.

We believe that our responses and the additional experiments have successfully addressed the concerns raised, and thank you again for your valuable feedback, which has directly contributed to strengthening this work.

---

### Decision · Program_Chairs · 2025-09-17

**Decision:**

Accept (poster)

**Comment:**

This paper aims to learn salient features for supervised learning tasks, separating out non-salient features into a subset of the feature space dimensions. Unlike previous methods, the proposed H-SPLID algorithm learns to separate these salient and non-salient features without additional data that goes beyond the data used for supervised learning. The loss function contains 3 types of terms. The cross entropy term is the cross entropy loss when predicting from only the salient features. The masked clustering terms encourage features to match mean features, with class-specific means for salient features and a global mean for non-salient features. Lastly, the HSIC terms minimize the dependence between the data and the salient features as well as the labels and the non-salient features. The H-SPLID algorithm minimizes the loss function through alternating optimization of the neural network and the saliency mask. H-SPLID is supported by a theoretical result that bounds the change in model prediction by the HSIC of the data and the salient features. Experiments show that the method improves robustness to adversarial attacks and transfer accuracy for saliency benchmarks. An ablation additionally shows that performance degrades when only a subset of the loss terms are used.

The reviewers praised the exposition, technical soundness, algorithmic novelty, and results of the paper. They raised various questions and asked for several experiments. The authors address these effectively in the rebuttal. In the end all reviewers were positive to paper acceptance although most expressed a relatively low confidence in their assessment.